# Preventing Trauma and Grief in Emergency and Critical Care Units: A Mixed Methods Study on a Psycho-Educational Defusing Intervention

**DOI:** 10.3390/healthcare12171800

**Published:** 2024-09-09

**Authors:** Francesco Tommasi, Paolo Tommasi, Marco Panato, Davide Cordioli, Riccardo Sartori

**Affiliations:** 1Department of Human Sciences, University of Verona, 37129 Verona, Italy; riccardo.sartori@univr.it; 2Azienda ULSS 9 Scaligera, 37122 Verona, Italy; paolo.tommasi@aulss9.veneto.it (P.T.); marco.panato@aulss9.veneto.it (M.P.); davide.cordioli@aulss9.veneto.it (D.C.)

**Keywords:** psycho-educational training, defusing, prevention programs, healthcare, mixed methods approach

## Abstract

Emergency and critical care services inevitably expose their staff to potential work stressors and traumatic events, which can cause emotional, behavioral, and physical reactions. The literature presents a wide range of evidence-based knowledge on the effectiveness of interventions to promote mental health after traumatic events. However, little is known about the effectiveness of prevention programs. In this study, we sought to improve the empirical understanding of the potential of a combination of psycho-educational-defusing training for trauma prevention. We employed a mixed methods approach using statistical modeling and content/focus group analysis to describe the sample of investigation and the effectiveness of the prevention training. A retrospective quantitative chart measured and evaluated the psychological state of physicians, nurses, and registered nurses (N = 222). A retrospective qualitative chart examined staff accounts of traumas and their coping strategies via autobiographies (n = 26). Prospective focus groups examined participants of the psycho-educational defusing intervention administered (n = 61). Findings revealed different forms of experiencing grief and trauma. Prospective analysis of the training effectiveness revealed favorable perceptions by participants. Results support the formal implementation of continuous prevention, building relational support, and coping strategies as keys to recovery and preventing traumas.

## 1. Introduction

Trauma and grief are common in sectors that inevitably expose their staff to continuous work stress and traumatic events, particularly in Emergency and Critical Care Units (ECCU) of the health sector. Unpredictability, overcrowding, and continuous confrontation with a broad range of patients with different and disparate diseases, while also confronting time pressures and engagement with a multiplicity of responsibilities, are unceasing distressing conditions. These can cause emotional, behavioral, and physical reactions. This is also the case when witnessing the death of patients, as it can be emotionally draining while also representing a risk factor for burnout. Taken together, the distressing characteristics of the environment and the exposure to traumatic incidents and death contribute to acute and chronic trauma that can build cumulatively over time [1,2,3]. It is unsurprising that clinical conditions, such as depression, anxiety disorder, and post-traumatic stress disorder, are prevalent among ECCUs, and they persist despite decades of knowledge and efforts to prevent them. By way of an example, meta-analytic investigations reported that one out of four nurses are at risk of developing a clinical condition, with similar trends before and after the pandemic [4,5,6,7]. Indubitably, reasons for this relate to these health services’ working and organizational characteristics.

To prevent clinical conditions of ECCU staff, the literature presents many evidence-based interventions to support such workers. Scholars and practitioners developed professional-oriented evidence-based training interventions following different approaches (i.e., cognitive behavioral therapy, CBT, and mindfulness approach [8], relational supportive [9], and coping mechanisms approach [10]). However, such initiatives often involve best practices for addressing specific clinical conditions (e.g., post-traumatic disorder) or propose only person-oriented strategies to help workers cope with their own experiences. However, the working and organizational context of ECCU can leverage the effectiveness of occasional person-oriented training initiatives [11,12,13,14]. Work stress and traumatic events may require the presence of mental-health experts conducting primary prevention programs in place (e.g., debriefing), which may not be present due to the lack of personnel in the unit [15]. Simultaneously, job demands and tasks may reduce the possibility of having the chance and the time to devote attention to the psychological states of the staff [5,16].

According to the literature, working and organizational conditions exposed to work stress and traumatic events may require initiatives to prevent and relational support [3,5,16,17]. Combining the lack of working and personnel resources with the constant exposure to stressful aspects, prevention initiatives can reduce the risk of clinical condition development. This can be the case of training interventions meant to promote psychological knowledge (e.g., psycho-educational interventions) or psychosocial competencies and peer-supporting (e.g., defusing) of the staff [11,13,14]. The former follows the evidence that individuals with psychological knowledge can better understand and identify emotions and feelings, especially after traumatic events [13,14]. The latter highlights the importance of building forms of relational support by fostering competencies for peer support. As for psycho-education, psychosocial competencies can help to act in situ directly after traumatic events and support the elaboration of cognitions, emotions, and feelings [14]. However, the potential of such prevention programs and whether they can effectively reduce the development of clinical conditions among the staff are still unclear [13,14]. The literature presents only 23 cases of training interventions for peer-supporting [18,19], while differences persist between countries. For example, in the Italian context, while the national regulation recognizes the importance of psychological support for ECCU staff, initiatives are sparse and not formally realized and supported on a national level [20]. Taken together, these aspects highlight the need to better inform the introduction of initiatives to prevent and support ECCU staff, especially in the case of specific structural barriers (e.g., lack of resources).

The present paper reports the results of a mixed methods study meant to evaluate an intervention involving a combination of psycho-educational and defusing training. The above insights provided the impetus to conceptualize trauma determinants and management. They suggested a feasible path to address the complex interplay between work stress (e.g., high exposition to stressful and traumatic events) and structural barriers (e.g., lack of resources) to clinical conditions prevention in the ECCU [18,19,21]. Considering the lack of organizational-based initiatives and permanent psychological health programs, our overreaching aim is to advance our empirical knowledge of the potential of prevention programs based on facilitating psychological knowledge and skills to deal with traumatic events in ECCU. Notably, we seek to complement the existing perspectives on pre by analyzing our intervention in order to inform the introduction of mental health prevention programs in the ECCU context. We do so by reporting the results of a mixed methods study. Following the call for the use of mixed methods to evaluate intervention [22], we used such an approach for purposes of (a) expansion (extending breadth and scope) to allow exploration of multiple levels of influence and (b) triangulation to assess the extent to which qualitative and quantitative findings corroborate each other [23]. As such, the present study can help scholars to better understand the effects of a specific prevention program while also supporting practitioners and healthcare managers in realizing similar initiatives.

## 2. Materials and Methods

### 2.1. Psycho-Educational-Defusing Training

Surprisingly, although the literature presents possibilities to integrate psycho-educational and defusing programs, there must be evidence of how such a combination could work in ECCU [18,19,21]. First, psycho-educational programs are meant to foster an understanding of the psychological aspects involved in work-related stress and traumatic events. The core idea is to educate the trainees to recognize the psychological valence in their work to foster their capacity to acknowledge their psychological health, risks for their health, and, more broadly, to talk about their psychological suffering. As such, this type of intervention involves transferring essential psychological pieces of knowledge followed by examples from concrete events [13].

Second, a defusing intervention program involves training specialists, i.e., defusers, to manage brief activities after traumatic situations. The defusing intervention follows the basic assumptions of a debriefing session where a mental health professional guides a group of people exposed to a traumatic incident in order to process it. That is, a defusing intervention sees peers trained to guide small groups of people to process traumatic incidents. However, defusers are not trained to perform the same functions as those specialized in mental health and do not replace them. The defusing approach has one main difference from other types of support intervention, namely the duration of the intervention: in fact, the work of the defusers is of short duration, aimed at processing the stressful and traumatic event according to a specific protocol. Moreover, it occurs immediately after the request by the beneficiaries and in situ. A defusing session includes an initial moment of gathering with purposes of (a) sharing and bringing out events, feelings, and thoughts related to the traumatic events, (b) initiating processing of them (i.e., events, feelings, and thoughts) with (c) the peer reassurance [14]. These elements are the object of an intervention aimed at training professionals as defusers and follow both theoretical and practical training.

Ultimately, while the psycho-educational programs offer the basis to open to the psychological dimension at work, training defusers is effective insofar as it aims to offer practical knowledge for on-time peer-support interventions after a traumatic event. Psycho-educational and defusing training can work in tandem to help improve both the knowledge and the skills of ECCU.

Following existing evidence-based perspectives while considering possible structural barriers and inevitable work-related stress in ECCU, we propose a unique intervention meant to restore and guarantee the well-being of ECCU staff that can be effective and timely in its realization by de-pathologizing and re-dimensioning traumatic events in situ [9,13,14,21]. We did so by proposing two separate training in on intervention session. While the first part was devoted to psycho-educational training (four hours), the second part aimed at offering the knowledge and practical skills for conducting a defusing session.

### 2.2. Evaluating the Intervention: Mixed Methods

To evaluate our intervention, we conducted a mixed methods project to obtain a retrospective and prospective picture of the state of the trainees and the quality of the training [23] (see Figure 1). Our mixed methods projects are theoretically driven by a deductive approach. As depicted in Figure 1, the first and the second study where represent the retrospective chart, i.e., the psychological status of the staff. In this, quantitative and qualitative data were collected concurrently but analyzed independently. Then, we merged the results and integrated them to create the research narrative and conduct the intervention. The last study was qualitative and represented the prospective component realized after the intervention in order to capture the experiences and perceptions of the effectiveness of the training among the participants.

In particular, to carry out the psychological support intervention, we conducted two retrospective studies to understand and highlight the actual incidence of the mental health of ECCU staff as well as to understand the valence of traumatic events. The two studies consisted of (a) a cross-sectional study based on self-report measures and (b) a qualitative study of autobiographies. Quantitative data were initially collected via an online questionnaire to collect evidence of work-related stress and states of distress. Qualitative data were collected utilizing autobiographical narratives to understand better how traumatic events occur in ECCU and how they can impact ECCU’s staff.

Second, the prospective component of our mixed methods study involved focus groups to understand the perspective of the ECCU’s staff on the intervention. The results of the retrospective component were initially given back to the participants. We continued with a discussion on the trauma and grief in ECCU and concluded by sharing perspectives on the interventions and their potential to reduce the development of clinical conditions.

### 2.3. Participants and Procedure

For the retrospective component concerning the cross-sectional study, 55% of the personnel of the ECCU (*N* = 404) of the five hospitals involved took part. The questionnaires were administered to 225 healthcare workers, of whom 222 consented to use the data. The target population consisted of 45 registered nurses, 136 nurses, and 41 physicians. The average age of the participants was 43 years (*M* = 42.6), and there was a prevalence of women corresponding to 70% of the sample (*n* = 157). Regarding the qualitative retrospective component, 21 participants belonged to the five previously mentioned hospitals. Lastly, 61 participants among the staff were involved in the prospective component. Participation in our project was voluntary, and there was not any financial incentive for it.

We began with the cross-sectional study by administrating an online questionnaire on ECCU’s staff from five local hospitals. Subsequently, autobiographical stories were collected by asking the same sample to realize an autobiography of their traumatic events. At the end of the retrospective part, we continued with three days of intervention with three different groups of ECCU staff belonging to the cross-sectional study. Each session lasted eight hours and did not include working hours, during which participants were introduced to the notions of clinical conditions such as psychological distress, anxiety, depression, and panic attack disorder and were invited to open up and collectively share their experience (i.e., psycho-educational program). During this session, the results narrative of the two retrospective studies was returned in order to offer a depiction of the psychological status of the staff. This allowed us to offer a concrete basis and examples for realizing the psycho-educational program. After the educational phase, we concluded each session with the defusing program. In this case, we explained defusing and simulated a session of defusing. During the simulation, trainees and participants role-played different possible events that an ECCU employee may witness during their shift (e.g., handling a relative in a crisis of anger). At the end of the simulations impersonated by the participants themselves, they were asked to conduct a defusing session by recounting the events, reporting thoughts and emotions, and trying to console themselves. During the simulations, it was ensured that the activity could be interrupted at any time by using a safe word that participants could use in the face of overly emotional states. Finally, we concluded each meeting by training volunteers to specialize in the defusing stages.

Finally, after the intervention days, we invited participants of the training to take part in follow-up meetings to discuss the quality of the training and their experiences after that (i.e., prospective study, focus groups). That is, two researchers realized three focus groups in which they opened the discussion among participants by asking them to report their experiences and perceptions of the training. No additional specific questions guided the discussion as space was offered for sharing thoughts in an open and inclusive climate. These elements were important for capturing the narratives of the participants whose analysis helped to understand the effectiveness of the intervention.

In all of the phases of our mixed methods study, participants were informed about the aim of the study and instructed about the procedure by the researchers. Informed consent was obtained from all participants whose data were used in the study, maintaining their anonymity. Moreover, given the content of the questions, participants could withdraw from each study whenever they wanted, and psychological support was offered upon request.

### 2.4. Instruments

In respect to the cross-sectional study, we employed the following self-report measures. To assess the level of work-related stress among ECCU staff, we used the Maslach Burnout Inventory (MBI). It comprises three components: Emotional Exhaustion (EE), Depersonalization Symptoms (DS), and Low Personal Accomplishment (LPA) symptoms (Maslach et al., 1997). We used the Italian version of the MBI adapted for the health sector [24,25] (Cronbach’s α = 0.86). We included a measure of death relation to evaluate how traumatic events could influence not only the level of work-related stress but also the way ECCU’s staff experience death. We used the Italian version of the Death Attitude Profile-Revised (DAP-R), i.e., a multidimensional scale relating to different attitudes toward death [26]. The DAP-R represents a broad spectrum of attitudes towards death, ranging from wanting to avoid it to having a neutral approach and finally accepting it. This scale proves to be functional and effective as it is also used in educating ECCU staff regarding the variety of attitudes one can have towards death (Cronbach’s α = 0.70). Finally, we assessed the quality of ECCU’s coping strategies. We followed Zimmer-Gembeck and Skinner’s (2011) approach, which allows us to identify personal strategies that are later categorized and considered. Participants report the way they approach a certain traumatic event and list a series of coping strategies. This series is later organized into coping strategies (i.e., problem-solving, comforting, distraction, escape, and information-seeking) [27].

Concerning the qualitative study of the retrospective components, we shared the following assignment via the institutional email of the five hospitals: “If you have ever had traumatic experiences (or something that you experienced as very emotional and impressive), please share it with us reporting what happened and what you have done.” In the email, we reported that we were looking for autobiographies, and we did not give any limitation to the number of words that they could use to tell their story.

Finally, we conducted three focus groups to follow up on the training sessions. We invited those who took part in our training by asking them to attend a meeting in which they could share (a) their impressions about the training and (b) their effectiveness. The three sessions started with researchers recalling the activities and lasted 70 min on average. We collected data via notes and recordings of the discussions.

### 2.5. Analytical Strategy

Our study is the first of its nature, so we employed an exploratory approach in our analysis. For the cross-sectional data, we limited to descriptive statistics. Data collected via autobiographies and focus groups were analyzed manually using content analysis.

### 2.6. Ethical Considerations

Given the themes of our project, the ethical consideration undertook three stages. First, we submitted our project to the Ethics Committee of the Department of Human Sciences (cf. Comitato Etico del Dipartimento di Scienze Umane), University of Verona. After the approval, the project was evaluated by the Ethical Committee of the Authors 2–4’ Hospital (cf. Comitato Etico Pratica Clinica). Finally, we concluded via the joint confirmation of the two Ethical Committees (cod. 2022_14) that our study’ method was performed in accordance with the Declaration of Helsinki.

## 3. Results

### 3.1. Retrospective: Quantitative

Descriptive statistics of the data collected showed that 40% of the participants scored high levels of burnout. Regarding the specific components of symptoms, 23% reported high levels of EE symptoms, with an increase in depersonalization symptoms for 62% of the sample and higher levels of PLA for 79% of the cases. Concerning how ECCU staff experience and think about death in their job, we found that 19% reported an avoidant relationship with the thought of death. In comparison, 10% of the sample has a cynical relationship with death. The fear of death is more present, with 28.3% reported being afraid of death for themselves and their loved ones, in addition to the patient’s death fear in 36.3% of cases. A high percentage of dysfunctional forms of coping strategies were reported by the participants, which appear to be in line with the results of burnout symptoms and death relations. Results show that only 18% of the participants employ problem-solving or comforting strategies (i.e., 16%). A high percentage of participants opted for the distraction and escaping approach, with 84.3% of the sample.

### 3.2. Retrospective: Qualitative

Content analysis of the autobiographies revealed that ECCU staff propose subjective accounts of trauma and grief and the way they try to overcome emotional, behavioral, and physical reactions. Notably, the content analysis revealed five main elements representing trauma and grief in the ECCU context and the way ECCU staff respond to traumatic incidents, namely (a) incidents are disparate, and trauma is cumulative, (b) response to traumatic incidents and (c) psychological suffering with (d) difficulties in recovering from traumatic incidents which result in feeling the (e) need for support.

First, (a) ECCU staff define traumatic incidents as presenting a wide variety of events that are also cumulative. Events are disparate, e.g., emergencies, random events, and death of a patient. Intriguingly, even ordinary events, e.g., structural stressing elements, can represent a traumatic event during a workday. It appears that ECCU staff consider both acute temporary stress and longstanding highlighting elements as traumatic. Moreover, the stories reported how (b) the ECCU staff responded to traumatic incidents. Notably, the stories show how they follow professional knowledge and invest in their work commitment to repress emotions and feelings after a traumatic event. This experience (c) results in an intense sense of grief characterized by low self-esteem and exhaustion. The incident broadens such feelings in the following days with senses of heaviness, melancholy, or a state of alertness in the subsequent period (e.g., weeks, months, or years). It is interesting to note that (d) participants reported how traumatic events were crystal clear in their memory despite years of working in the same area, with some participants reporting suffering clinical conditions (e.g., chronic panic attack disorder). Lastly, our story analysis showed (e) how participants felt they needed support. This has been reported in stories relating to specific cases of acute stress as well as by those who presented structural stress elements, claiming the impossibility of dealing with their grief.

### 3.3. Prospective: Quantitative

Results from the three focus groups led to the identification of two main elements of our intervention, namely, (a) the effectiveness and (b) the limitations of the training. Firstly, participants reported how they spent more time reflecting on their daily work experience by noting thoughts and feelings related to their tasks and emotionally impactful events. Some of them reported how they had also had experiences of defusing with a small group of colleagues meeting after the shift to reconstruct the workday and reduce its emotional involvement. In other cases, informal defusing meetings happened during breaks, in which they brought out incidents, feelings, and thoughts related to a stressful task during the workday for processing it in a peer-supportive context. Together, these experiences demonstrated the effectiveness of the training, with participants reporting a better quality of life. For example, participants showed more awareness of their feelings and thoughts (e.g., negative emotions, fears, insomnia). Recalling the words of one participant, the training allowed them to “breathe” and “reconnect with themselves and their coworkers in the daily pressure of the emergency department.” Unsurprisingly, the participants benefited from defusing as characterized by its timely occurrence and the relational aspects among peers.

Concerning effectiveness, some participants reported how the training activity made aspects of themselves re-emerge. In one group, participants lived a traumatic event with a colleague of them committing suicide. This resulted in prolonged acute stress within the group, who reported the need for additional psycho-educational training to support them. In their words, this is not one limiting aspect of the training activity as it helped to elaborate their trauma and live their grief with a more precise vocabulary that allowed them to express their emotions and thoughts about the event. Moreover, this group and other participants reported how the training helped them feel more human and less like a group of ECCU personnel, considering the human side a crucial part of their work.

Despite this, participants were reminded of the structural barriers that limit their capacity for relational support. The ECCU is characterized by the exposition to acute stressful events in which there is not always space for caring for each other and sharing lived experiences. In this, it appeared that the training activity is only one step that should represent a way forward for improving the working conditions to allow more “human space” beyond the inevitable acute stress of ECCU. For this reason, participants asked for additional psycho-educational interventions to advance their knowledge and skills for themselves and others.

## 4. Discussion

In this article, we reported the results of a mixed-method project aimed at evaluating a psycho-educational-defusing intervention for preventing trauma and grief in ECCU’s staff. Considering the ambition of the study, we followed the mixed-method approach for (a) expansion to allow exploration of multiple levels of influence and (b) triangulation to assess the extent to which qualitative and quantitative findings corroborate each other [23]. The strength of this methodology stands in the qualitative findings, expanding the understanding and uncovering possible explanations for quantitative findings. This is the case for the autobiographies of traumatic events by which we are able to understand the causes of the persistent clinical conditions among the participants. Moreover, the prospective qualitative data of the focus group allowed us to understand the effectiveness of the training. Indeed, to evaluate the effectiveness of defusing sessions, quantitative methods are limited as defusing sessions occur occasionally and based on individual needs. Likewise, prospective quantitative studies would be limited as aspects that emerged from the qualitative sessions cannot be operationalized. However, our methodology and results may inform subsequent studies in which statistical models can be realized to assess the effectiveness of psycho-educational-defusing interventions.

Our mixed-method project found that psycho-educational-defusing interventions can be effective in preventing traumatic events despite the resource limitations of ECCU. The units of emergency and critical care in the healthcare sector inevitably expose their staff to acute stress and traumatic events. Moreover, these units do not always present resources for preventing clinical conditions and promoting the well-being of their team. Considering these barriers, our psycho-educational-defusing intervention appeared as a flexible and practical approach against organizational obstacles. As such, our study responds to the call for realizing empirical investigations of prevention programs [19] while also providing the first evidence of the effectiveness of combined training [9,13,14,21]. Traumas and grief among ECCUs can likely persist unless psychological knowledge and psychosocial competencies are addressed. Results of our project revealed that (a) improving knowledge and competencies to express feelings and thoughts related to stressful events while (b) creating the conditions for peer-supporting may be even more effective than secondary prevention programs [9,13,14,19,21].

According to the literature, ECCU staff are constantly exposed to potentially traumatic incidents with a high risk of emotional, behavioral, and physical reactions and the development of clinical conditions. This study’s results were not an exception, and we found similar trends of psychological suffering among ECCU staff due to traumatic incidents and distressing conditions [1,2,3,4,5,6,7]. In this, trauma is cumulative with a sequela of psychological suffering that affects the overall experience and practice of the staff [3]. That is, trauma and grief appear as emotional, behavioral, and physical reactions that result from unpredictability, overcrowding, and continuous confrontation with a broad range of traumatic incidents. However, this is not only related to the momentary, occasional experience but last over time, which can have a sequela of psychological suffering that impacts the way (a) ECCU staff are able to overcome distressing conditions and (b) their practices. In this, supplying ECCU staff with group resources may be beneficial and create the conditions for empowering ECCU staff [28]. Results of the focus groups revealed that the main contribution of our intervention lies in the provision of psychological knowledge, which offers a vocabulary to the staff for verbalizing and sharing their inner thoughts and emotions while also creating opportunities for recognizing their grief. In parallel, peer support appeared to be a tool for overcoming distressing incidents as participants of the focus group reported how they had occasion to reduce their emotional fatigue by engaging in defusing sessions [28,29]. This resulted to be helpful for processing the incident but also to build meaningful relationships. Taken together, these elements indicate that our training, while offering an essential resource for ECCU staff, also supports the quality of teamwork and relationships among peers.

### 4.1. Limitations and Future Research

This study is the first of its kind to examine primary prevention programs via a mixed method project, and as such, it is intended as exploratory. While this can limit its results, it also yields a series of implications for future research. First, our project participants were from a specific context (i.e., the Italian health sector) and specific healthcare sector departments (i.e., ECCU); they may not reflect the broader population. Despite this, our results represent a basis for future exploration extending to other countries and departments. Second, the absence of quantitative evidence of the effectiveness of the study requires careful consideration. Future investigations can consider implementing statistical modeling to reflect the vast experiences of this staff. Longitudinal studies and experimental studies involving control groups can be implemented to provide evidence-based knowledge in support of our findings. Lastly, our qualitative studies involved only a small proportion of the ECCU staff of the hospitals involved, and no physicians attended the focus groups. This suggests that there might be an additional number of different meanings that ECCU staff use to talk about trauma and grief. However, the variety of meanings reported indicates that we must also consider with caution any research that might stereotype the lived experience of individuals.

In addition, future studies can also take into account our results independently of our study limitations. Notably, our results suggest several implications for the study of trauma and grief in the healthcare context. First, the fact that we identify a series of structural problems suggests that healthcare organizations should prioritize attention to trauma in the ECCU context. This can result in continuous monitoring of the staff while maintaining and creating occasions for psycho-education. Second, our results of the lived experience of ECCU staff appear to be ideal examples for understanding the disparate and different meanings of trauma and grief. These entail something more than objective traumatic acute stressful incidents. Our results show that trauma is cumulative [3] and that trauma results in repetitive and constant frustration of the staff, which leaves them less willing to express emotions and feelings [1,2,18]. Lastly, the present study adds to the current understanding of recovery from distressing events. While building resilience, developing coping strategies, and seeking professional support are essential keys to recovery, our study emphasizes the importance of community and peer support within the work environment. Soliciting peer support involves recognizing structural conditions while stressing the importance of peer relations and trust as reassuring resources that can foster recovery [28,29].

### 4.2. Practical Implications

Our results provided initial knowledge on the potential of psycho-educational defusing intervention as an effective training program for preventing mental-ill health problems among ECCU staff. Then, our results can help healthcare managers realize initiatives for providing psychological support to their staff, especially in the presence of a lack of financial resources for continuous support by mental health experts. That is, our study sharpens the specific role of psychological education and peer support. While psycho-education can foster awareness of the psychological effects of trauma and distressing conditions, defusing emphasizes the value of peer support by exchanging and sharing thoughts and feeling among colleagues in situ, right after incidents. In this, the main strength is the accomplished effectiveness of a relatively short and on-time intervention.

Realizing initiatives to mitigate the effects of the environment is important to improve the overall working experience of ECCU staff while also promoting the ECCU context itself and the patients for whom it cares. Accordingly, offering a supporting environment with an organizational network offers immediate defusing thanks to peer support [3,15]. In our study results, we showed how ECCU staff are in need of such types of intervention as they can be not only helpful but also necessary, with the addition of organizational support.

## 5. Conclusions

The present study, which adopted a mixed methods approach, explored the effectiveness of a psycho-educational-defusing intervention for preventing trauma and grief among the ECCU staff. Considering the continuous exposition to distressing conditions and traumatic incidents, ECCU staff confront a series of structural barriers that remind us of the need for in situ prevention programs. Our mixed methods study allowed us to find that trauma in the ECCU occurs in a variety of disparate ways and is cumulative. Trauma exerts effects on mental health and professional practice while also remaining unprocessed by the staff. Psycho-educational-defusing intervention can be a resource for ECCU departments as results showed that our training represented a protective factor thanks to the offer of knowledge and skills for coping with distress and trauma.

## Figures and Tables

**Figure 1 healthcare-12-01800-f001:**
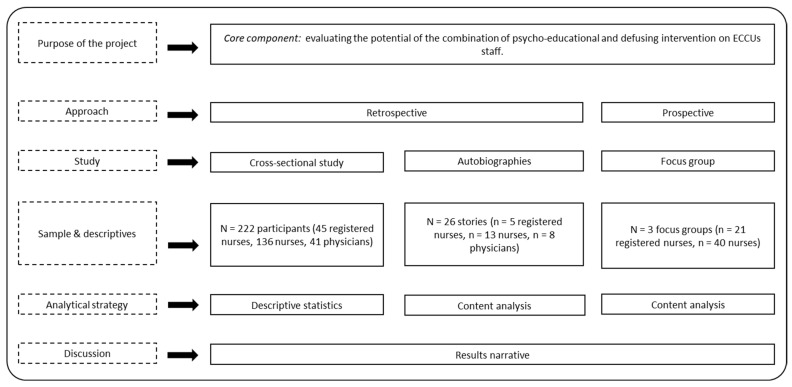
Graphical depiction of the mixed methods project.

## Data Availability

Quantitative and qualitative data are available upon request by the corresponding author.

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
