# Peer review of "Preventing Trauma and Grief in Emergency and Critical Care Units: A Mixed Methods Study on a Psycho-Educational Defusing Intervention"

_healthcare, 2024, doi:10.3390/healthcare12171800_

Round 1

Reviewer 1 Report

Comments and Suggestions for Authors

Overall, the article provides a valuable contribution to the field of organizational psychology in healthcare contexts, highlighting the importance of trauma and grief prevention. The sections are well-structured and informative but could be improved with a more critical analysis of the limitations and a clearer integration of qualitative and quantitative data.

The introduction effectively establishes the context and relevance of the study, mentioning the prevalence of conditions such as depression and post-traumatic stress disorder among ECCU workers. It provides an adequate literature review of existing interventions but could benefit from a more detailed discussion of the specific gaps the study aims to fill. The introduction could also be more explicit about the study's hypotheses and specific objectives.

The methods section is detailed and clearly describes the study design, participants, procedures, and instruments used. The description of the psychoeducational and defusing training is comprehensible, but details about the duration and frequency of the intervention sessions are missing. The choice of a mixed-methods approach is justified, but the explanation of how qualitative and quantitative data were integrated could be more explicit.

The results are presented in a structured manner, with a clear distinction between quantitative and qualitative analyses. However, the quantitative section could benefit from more robust statistical analysis and a more detailed discussion of the results concerning the initial hypotheses. The qualitative analysis offers valuable insights into participants' subjective experiences, but the data categorization could be more detailed.

The discussion interprets the results well, relating them to the existing literature and highlighting the potential effectiveness of combined interventions. However, it lacks critical analysis of the study's limitations, such as the sample size and the absence of a control group. The discussion could further explore practical implications and suggest clear recommendations for implementing interventions in other healthcare units.

The conclusion adequately summarizes the study's main findings and implications. However, it could be more assertive regarding recommendations for future research and clinical practices. The conclusion could also more explicitly address how the study's limitations impact the results and suggest improvements for future studies.

Author Response

RC1: Overall, the article provides a valuable contribution to the field of organizational psychology in healthcare contexts, highlighting the importance of trauma and grief prevention. The sections are well-structured and informative but could be improved with a more critical analysis of the limitations and a clearer integration of qualitative and quantitative data.

AR1: We wish to thank you for your appreciation. We are happy to read your positive words about our manuscript. Thanks! We have followed your constructive suggestions and comments which helped us to improve our manuscript. Please refer to the following responses and extracts from the manuscript text.

RC2: The introduction effectively establishes the context and relevance of the study, mentioning the prevalence of conditions such as depression and post-traumatic stress disorder among ECCU workers. It provides an adequate literature review of existing interventions but could benefit from a more detailed discussion of the specific gaps the study aims to fill. The introduction could also be more explicit about the study's hypotheses and specific objectives.

AR2: Thanks a lot for your suggestions which is also in line with the comments by reviewers 2 and 3. We revised the introduction section with a more explicit description of our objectives and purposes. You can find them in the manuscript text as well as in the following extract that we report for your convenience:

“The present paper reports the results of a mixed methods study meant to evaluate an intervention involving a combination of psycho-educational and defusing trainings. The above insights provided the impetus to conceptualize trauma determinants and management. They suggested a feasible path to address the complex interplay between work stress (e.g., high exposition to stressful and traumatic events) and structural barriers (e.g., lack of resources) to clinical conditions prevention in the ECCU [18,19,21]. Considering the lack of organizational-based initiatives and permanent psychological health programs, our overreaching aim is to advance our empirical knowledge of the potential of prevention programs based on facilitating psychological knowledge and skills to deal with traumatic events in ECCU. Notably, we seek to complement the existing perspectives on pre by analyzing our intervention in order to inform the introduction of mental health prevention programs in the ECCU context. We do so by reporting the results of a mixed methods study. Following the call for the use of mixed methods to evaluate intervention [22], we used such an approach for purposes of (a) expansion (extending breadth and scope) to allow exploration of multiple levels of influence and (b) triangulation to assess the extent to which qualitative and quantitative findings corroborate each other [23]. As such, the present study can help scholars to better understand the effects of a specific prevention program while also support practitioners and healthcare managers in realizing similar initiatives.”

RC3: The methods section is detailed and clearly describes the study design, participants, procedures, and instruments used. The description of the psychoeducational and defusing training is comprehensible, but details about the duration and frequency of the intervention sessions are missing. The choice of a mixed-methods approach is justified, but the explanation of how qualitative and quantitative data were integrated could be more explicit.

AR3: Thanks for your comment which is also in line with the comments by reviewers 2 and 3. We have clarified the way we interpreted the results and how we integrated them in our narrative results. See the following extract:

“Surprisingly, although the literature presents possibilities to integrate psycho-educational and defusing programs, there must be evidence of how such a combination could work in ECCU [18,19,21]. First, psycho-educational programs are meant to foster an understanding of the psychological aspects involved in work-related stress and traumatic events. The core idea is to educate the trainees to recognize the psychological valence in their work to foster their capacity to acknowledge their psychological health, risks for their health, and, more broadly, to talk about their psychological suffering. As such, this type of intervention involves transferring essential psychological pieces of knowledge followed by examples from concrete events [13].

Second, a defusing intervention program involves training specialists, i.e., defusers, to manage brief activities after traumatic situations. The defusing intervention follows the basic assumptions of a debriefing session where a mental health professional guide a group of people exposed to a traumatic incident in order to process it. That is, a defusing intervention sees peers trained to guide small groups of people to process traumatic incidents. However, defusers are not trained to perform the same functions as those specialized in mental health and do not replace them. The defusing approach has one main difference from other types of support intervention, namely the duration of the intervention: in fact, the work of the defusers is of short duration, aimed at processing the stressful and traumatic event according to a specific protocol. Moreover, it occurs immediately after the request by the beneficiaries and in situ. A defusing session includes an initial moment of gathering with purposes of a) sharing and bringing out events, feelings, and thoughts related to the traumatic events, b) initiating processing of them (i.e., events, feelings, and thoughts) with c) the peer reassurance[14]. These elements are the object of an intervention aimed at training professionals as defusers and follows both theoretical and practical training.

Ultimately, while the psycho-educational programs offer the basis to open to the psychological dimension at work, training defusers is effective insofar as it aims to offer practical knowledge for on time peer-support interventions after a traumatic event. Psycho-educational and defusing training can work in tandem to help improve both the knowledge and the skills of ECCU.

Following existing evidence-based perspectives while considering possible structural barriers and inevitable work-related stress in ECCU, we propose a unique intervention meant to restore and guarantee the well-being of ECCU staff that can be effective and timely in its realization by de-pathologizing and re-dimensioning traumatic events in situ [9,13,14,21]. We did so by proposing two separate training in on intervention session. While the first part was devoted to psycho-educational training (four hours), the second part aimed at offering the knowledge and practical skills for conducting a defusing session.”

RC4: The results are presented in a structured manner, with a clear distinction between quantitative and qualitative analyses. However, the quantitative section could benefit from more robust statistical analysis and a more detailed discussion of the results concerning the initial hypotheses. The qualitative analysis offers valuable insights into participants' subjective experiences, but the data categorization could be more detailed.

AR4: We really appreciate your comments. We have revised the result section concerning our qualitative results by emphasizing the categories emerged from our qualitative data analysis. While you can easily identify our revisions in the manuscript text, you can also refer to the following extract:

3.2. Retrospective: qualitative

Content analysis of the autobiographies revealed that ECCU's staff propose subjective accounts of trauma and grief and the way they try to overcome emotional, behavioural and physical reactions. Notably, content analysis revealed five main elements representing trauma and grief in ECCU context and the way ECCU staff respond to traumatic incidents, namely a) incidents are disparate and trauma is cumulative, b) response to traumatic incidents and c) psychological suffering with d) difficulties in recovering from traumatic incidents which result in feeling the e) need for support.

Firs, a) ECCU staff define traumatic incidents presenting a wide variety of events which are also cumulative. Events are disparate, e.g., emergencies, random events, death of a patient. Intriguingly, even ordinary events, e.g., structural stressing elements, can represent a traumatic event during a workday. It appears that ECCU's staff consider both acute temporary stress and longstanding highlighting elements as traumatic. Moreover, the stories reported how b) the ECCU's staff respond to traumatic incidents. Notably, the stories show how they follow professional knowledge and invest in their work commitment to repress emotions and feelings after a traumatic event. This experience c) results in an intense sense of grief characterized by low self-esteem and exhaustion. The incident broadens such feelings in the following days with senses of heaviness, melancholy, or a state of alertness in the subsequent period (e.g., weeks, months, or years). It is interesting to note that d) participants reported how traumatic events were crystal clear in their memory despite years of working in the same area, with some participants reporting suffering clinical conditions (e.g., chronic panic attack disorder). Lastly, our story analysis showed e) how participants felt they needed support. This has been reported in stories relating to specific cases of acute stress as well as by those who presented structural stress elements, claiming the impossibility of dealing with their grief.

3.3. Prospective: quantitative

Results from the three focus groups led to the identification of two main elements of our intervention, namely, a) the effectiveness and b) limitations of the training. Firstly, participants reported how they spent more time reflecting on their daily work experience by noting thoughts and feelings related to their tasks and emotionally impactful events. Some of them reported how they had also had experiences of defusing with a small group of colleagues meeting after the shift to reconstruct the workday and reduce its emotional involvement. In other cases, informal defusing meetings happened during breaks, in which they brought out incidents, feelings, and thoughts related to a stressful task during the workday for processing it in a peer-supportive context. Together, these experiences demonstrated the effectiveness of the training, with participants reporting a better quality of life. For example, participants showed more awareness of their feelings and thoughts (e.g., negative emotions, fears, insomnia). Recalling the words of one participant, the training allowed them to "breathe" and "reconnect with themselves and their coworkers in the daily pressure of the emergency department." Unsurprisingly, the participants benefited from defusing as characterized by its timely occurrence and the relational aspects among peers.”

RC5: The discussion interprets the results well, relating them to the existing literature and highlighting the potential effectiveness of combined interventions. However, it lacks critical analysis of the study's limitations, such as the sample size and the absence of a control group. The discussion could further explore practical implications and suggest clear recommendations for implementing interventions in other healthcare units.

AR5: We agree with your points which are also in line with the points made by reviewers 2 and 3. Thanks for your suggestions to make our analysis more critical concerning our limitations and future research perspectives. We have operated in the manuscript text and we revised the entire discussion as follows:

  1. Discussion

In this article, we reported the results of a mixed-method project aimed at evaluating a psycho-educational-defusing intervention for preventing trauma and grief in ECCU's staff. Considering the ambition of the study, we followed the mixed-method approach for (a) expansion to allow exploration of multiple levels of influence and (b) triangulation to assess the extent to which qualitative and quantitative findings corroborate each other [23]. The strength of this methodology stands in the qualitative findings, expanding the understanding and uncovering possible explanations for quantitative findings. This is the case for the autobiographies of traumatic events by which we are able to understand the causes of the persistent clinical conditions among the participants. Moreover, the prospective qualitative data of the focus group allowed us to understand the effectiveness of the training. Indeed, to evaluate the effectiveness of defusing sessions, quantitative methods are limited as defusing sessions occur occasionally and based on individual needs. Likewise, prospective quantitative studies would be limited as aspects that emerged from the qualitative sessions cannot be operationalized. However, our methodology and results may inform subsequent studies in which statistical models can be realized to assess the effectiveness of psycho-educational-defusing interventions.

       Our mixed-method project found that psycho-educational-defusing interventions can be effective in preventing traumatic events despite the resource limitations of ECCU. The units of emergency and critical care in the healthcare sector inevitably expose their staff to acute stress and traumatic events. Moreover, these units do not always present resources for preventing clinical conditions and promoting the well-being of their team. Considering these barriers, our psycho-educational-defusing intervention appeared as a flexible and practical approach against organizational obstacles. As such, our study responds to the call for realizing empirical investigations of preventing programs [19], while also providing the first evidence of the effectiveness of a combined training [9,13,14,21]. Traumas and grief among ECCUs can likely persist unless psychological knowledge and psychosocial competences are addressed. Results of our project revealed that a) improving knowledge and competences to express feelings and thoughts related to stressful events while b) creating the conditions for peer-supporting maybe even more effective than secondary prevention programs [9,13,14,19,21].

According to the literature, ECCU staff are constantly exposed to potential traumatic incidents with a high risk for their staff for emotional, behavioural and physical reactions and the development of clinical conditions. This study’s results were not an exception and we found similar trends of psychological suffering among ECCU staff due traumatic incidents and distressing conditions [1–7]. In this, trauma is cumulative with a sequela of psychological suffering that affects the overall experience and practice of the staff [3]. That is, trauma and grief appear as emotional, behavioural, and physical reactions that result from the unpredictability, overcrowding, continuous confrontation with a broad range of traumatic incidents. However, this is not only related to the momentary, occasional experience but last overtime which can have a sequela of psychological suffering that impacts the way a) ECCU’s staff are able to overcome distressing conditions and b) their practices. In this, supplying ECCU staff with group resources may be beneficial and create the conditions for empowering ECCU staff [28]. Results of the focus groups revealed that the main contribution of our intervention lays in the provision of psychological knowledge which offers a vocabulary to the staff for verbalizing and sharing their inner thoughts and emotions while also creating opportunities for recognizing their grief. In parallel, peer-support appeared to be a tool for overcoming distressing incident as participants of the focus group reported how they had occasion to reduce their emotional fatigue by engaging in defusing sessions [28,29]. This resulted to be helpful for processing the incident but also to build meaningful relationships. Taken together, these elements indicate that our training while offering a essential resource for ECCU staff, it also support the quality of team work and relationships among peers.

4.1 Limitations & Future research

This study is the first of its kind to examine primary preventing programs via a mixed method project, and as such, our is intended as exploratory. While this can limit its results, it also yields a series of implications for future research. First, our project participants were from a specific context (i.e., the Italian health sector) and specific healthcare sector departments (i.e., ECCU) they may not reflect the broader population. Despite this, our results represent a basis for future exploration extending to other countries and departments. Second, the absence of quantitative evidence of the effectiveness of the study requires careful consideration. Future investigations can consider implementing statistical modeling to reflect the vast experiences of this staff. Longitudinal studies and experimental studies involving control groups can be implemented for providing evidence-based knowledge in support to our findings. Lastly, our qualitative studies involved only a less proportion of the ECCU staff of the hospitals involved and no physicians attended the focus groups. This suggests that there might be an additional number of different meanings that ECCU staff use to talk about trauma and grief. However, the variety of meanings reported indicates that we must also consider with caution any research that might stereotype the lived experience of individuals.

In addition, future studies can also take into account our results independently of our study limitations. Notably, our results suggest several implications for the study of trauma and grief in the healthcare context. First, the fact that we identify a series of structural problems suggests that healthcare organizations should prioritize attention to trauma in the ECCU context. This can result in continuous monitoring of the staff while maintaining and creating occasions for psycho-education. Second, our results of the lived experience of ECCU staff appear to be ideal examples for understanding the disparate and different meanings of trauma and grief. These entail something more than objective traumatic acute stressful incidents. Our results show that trauma is cumulative [3] and that trauma results in repetitive and constant frustration of the staff, which leaves them less willing to express emotions and feelings [1,2,18]. Lastly, the present study adds to the current understanding of recovery from distressing events. While building resilience, developing coping strategies, and seeking professional support are essential keys to recovery, our study emphasizes the importance of community and peer support within the work environment. Soliciting peer support involves recognizing structural conditions while stressing the importance of peer relations and trust as reassuring resources that can foster recovery [28,29].

4.2 Practical implications

Our results provided initial knowledge on the potential of psycho-educational defusing intervention as an effective training program for preventing mental-ill health problems among ECCU staff. Then, our results can help healthcare managers in realizing initiatives for providing psychological support to their staff, and especially in presence of lack of financial resources for continuous support by mental health experts. That is, our study sharpens the specific role of psychological education and peer support. While psycho-education can foster awareness over the psychological effects of trauma and distressing conditions, defusing emphasizes the value of peer supporting by exchanging and sharing thoughts and feeling among colleagues in situ, right after incidents. In this, the main strength is the accomplished effectiveness o f a relatively short and on time intervention.

Realizing initiatives to mitigate the effects of the environment is important to improve the overall working experience of ECCU staff while also promoting the ECCU context itself and the patients for which it cares of. Accordingly, offering a supporting environment with an organizational network offering immediate defusing thanks to peer support [3,15]. In our study results, we showed how ECCU staff are in need for such types of intervention as they can be not only be helpful but also necessary, with the addition of organizational support.

CR6: The conclusion adequately summarizes the study's main findings and implications. However, it could be more assertive regarding recommendations for future research and clinical practices. The conclusion could also more explicitly address how the study's limitations impact the results and suggest improvements for future studies.

AR6: Thanks for your comment. Due to changes in the discussion section, we re-wrote the entire conclusion of our manuscript. We did so also to be more explicit about our contributions. Please, refer to the following extract:

  1. Conclusion

               The present study, which adopted a mixed methods approach, explored the effectiveness of a psycho-educational-defusing intervention for preventing trauma and grief among the ECCU staff. Considering the continuous exposition to distressing conditions and traumatic incidents, the ECCU staff confronts with a series of structural barriers which remind to the need for in situ prevention programs. Our mixed methods study allowed us to find that trauma in the ECCU occurs in a variety of disparate ways and is cumulative. Trauma exerts effects on the mental health, and professional practice while also remaining unprocessed by the staff. Psycho-educational-defusing intervention can be a resource for ECCU departments as results showed that our training represented a protective factor thanks to the offer of knowledge and skills for coping with distress and trauma.

Reviewer 2 Report

Comments and Suggestions for Authors

   Preventing Trauma and Grief in Emergency and Critical Care Units: A Mixed Methods Study on a Psycho-educational Defusing Intervention

Abstract

The abstract is very good. I would appreciate it if you could explain how you measured resilience.

1.      Introduction

Dear authors, first of all thank you for the article. The content is very interesting. Dealing with aspects related to Emergency and Critical Care Units (ECCU) is of vital importance.

In the introduction it would be important to define trauma and grief. Many readers will not be familiar with these two concepts. Exactly the same with post-traumatic stress disorder.

¿At this time there are no formal psychological support programs in ECCU?. Also, it would be important to know if this lack of support happens equally in emerging and developed countries. It is reasonable to think that in emerging countries the pressure of time and the multiplicity of roles is more evident in ECCU.

In the introduction they should contextualize the problem. The introduction is too general, and I am convinced that the lack of labor and personal resources is different in each country. Therefore, the strategies they propose could go in a more concrete direction.

Psycho-educational interventions or self-knowledge to identify emotions together with relational resilience are adequate prevention strategies. A mixed study is interesting because it is unusual. However, the introduction should be improved by taking into account all the above points. At the moment it is too general and may confuse the reader.

Figure 1 (method used) would need further explanation. It could also be placed in another location.

2.      Materials and Methods

2.1. Psycho-educational-defusing training

There is no article combining psycho-educational and defusing programs (effect on ECCU)?, that aspect is very relevant.

The role and relevance of psycho-educational programs is adequately explained. The role and relevance of defusing programs is also well explained. The priority focus of the defusing program is well understood.

However, the impact that defusing programs and psycho-educational training can have on ECCU can be improved. In this way it is possible to highlight the content of your article with respect to previous ones. Furthermore, to emphasize the gap in knowledge about these relationships.

This section in the methods area is somewhat strange. Reconsider its location in the introduction.

2.2. Evaluating the intervention: Mixed methods

Possibly Figure 1 would fit better in this section.

The mixed study configuration is well explained. Moreover, it is one of the relevant parts of this article.

An explanation of the online questionnaire (process) is needed. What guarantees did the participant have, e.g., confidentiality of data, voluntary withdrawal, consent to participate and so on. This information could be entered in the procedure section because I see that in this section they enter some data.

The results of the retrospective component were initially returned to the participants. Why?

2.3.  Participants & Procedure

Have the distribution of the population studied as a possible limitation of your article. 70% female.

Remember to contextualize the problem in the introduction and include more information on the five hospitals analyzed.  The qualitative retrospective component is well explained as is the whole process in general. Good job.

2.4. Instruments

Why did they use the Maslach Burnout Inventory (MBI) to measure stress? Possibly there are better and more specific instruments available. Using the Italian version is fine.

Using a measurement scale to assess how traumatic events might influence not only the level of stress is fine (Italian version of the Death Attitude Profile-Revised (DAP-R)).

The model of Zimmer-Gembeck and Skinner (2011) to identify personal strategies is very good.

The way to measure the qualitative part, specifically, the retrospective components is well explained.

They can explain in more detail the focus groups and their importance?.

2.5. Analytical strategy

This part is a priority, and the analysis is well focused. It is key to highlight that this is the first study of this nature (mixed). They did not use any software to analyze quantitative data, e.g., Atlas?.

2.6. Ethical considerations

Having the support of an ethics committee is of vital importance and enhances the results obtained.

3.      Results

3.1. Retrospective: quantitative

It is necessary to better explain the percentages obtained through descriptive statistics. That is, did you use correlations? Please detail all the processes used.

Specifying dysfunctional coping strategies is very important. This part is fine.

3.2. Retrospective: qualitative

I especially like the way they explain this part. Good job.

3.3. Prospective: quantitative

In the end there were six or three focus groups. This part is confusing.

In general, the prospective: quantitative is well explained.

4.      Discussion

The explanation of the strengths of the applied methodology is good. I agree on the importance of your study and that the mixed methodology is one of its main strengths. However, it is important to better discuss your findings and generate some kind of comparison with previous studies even if partial before proposing future research.

Being able to better analyze the results through discussion is key.

Their research suggests a greater number of practical applications. Expand on that part. I believe it is one of the main contributions of this research.

Check how you measured resilience.

Finally, it is essential to include limitations.

5.      Conclusion

The conclusion includes limitations and even future research. I suggest that you reconsider the discussion section by improving the practical implications, future research and including limitations. The conclusion section, in my opinion, should be redrafted through the main objective of this research and its possible fulfillment.

Author Response

Reviewer 2

RC: Abstract. The abstract is very good. I would appreciate it if you could explain how you measured resilience.

AR: Thanks for your comment, we appreciate that you liked our abstract. Indeed, we did not measure resilience. To avoid confusion, we deleted the word “resilience” and limited to describe the dimensions considered in our study. For your convenience, we report the abstract below:

“Emergency and critical care services inevitably expose their staff to potential work stressors and traumatic events, which can cause emotional, behavioral, and physical reactions. The literature presents a wide range of evidence-based knowledge on the effectiveness of interventions to promote mental health after traumatic events. However, little is known about the effectiveness of prevention programs. In this study, we sought to improve the empirical understanding of the potential of a combination of psycho-educational-defusing training for trauma prevention. We employed a mixed methods approach using statistical modeling and content/focus group analysis to describe the sample of investigation and the effectiveness of the prevention training. A retrospective quantitative chart measured and evaluated the psychological state of physicians, nurses, and registered nurses (N = 222). A retrospective qualitative chart examined staff accounts of traumas and their coping strategies via autobiographies (n = 26). Prospective focus groups examined participants of the psycho-educational defusing intervention administered (n = 61). Findings revealed different forms of experiencing grief and trauma. Prospective analysis of the training effectiveness revealed fa-vorable perceptions by participants. Results support the formal implementation of continuous prevention, building relational support, and coping strategies as keys to recovery and preventing traumas.”

RC1: Introduction. Dear authors, first of all thank you for the article. The content is very interesting. Dealing with aspects related to Emergency and Critical Care Units (ECCU) is of vital importance. In the introduction it would be important to define trauma and grief. Many readers will not be familiar with these two concepts. Exactly the same with post-traumatic stress disorder. ¿At this time there are no formal psychological support programs in ECCU?. Also, it would be important to know if this lack of support happens equally in emerging and developed countries. It is reasonable to think that in emerging countries the pressure of time and the multiplicity of roles is more evident in ECCU. In the introduction they should contextualize the problem. The introduction is too general, and I am convinced that the lack of labor and personal resources is different in each country. Therefore, the strategies they propose could go in a more concrete direction. Psycho-educational interventions or self-knowledge to identify emotions together with relational resilience are adequate prevention strategies. A mixed study is interesting because it is unusual. However, the introduction should be improved by taking into account all the above points. At the moment it is too general and may confuse the reader.

Figure 1 (method used) would need further explanation. It could also be placed in another location.

AR1: Thanks for your positive words, we are happy that you enjoyed our manuscript and its contents. We wish to thank you for giving us the opportunity to explain in more details what we mean with trauma and grief. This is in line with the point made by reviewer 3. Moreover, we have explained the context of our study. Taken together, these elements helped us to improve the discussion of our manuscript. Finally, we moved Figure 1 to the Method section and provided additional explanation. We wish to thank you for your constructive comments as they helped us to improve the entire manuscript. For your convenience, we report the extract of our introduction below:

“1. Introduction

Trauma and grief are common in sectors that inevitably expose their staff to continuous work stress and traumatic events, particularly in Emergency and Critical Care Units (ECCU) of the health sector. Unpredictability, overcrowding, continuous confrontation with a broad range of patients with different and disparate of disease, while also confronting with time pressures and engagement with a multiplicity of responsibilities are unceasing distressing conditions. These can cause emotional, behavioural and physical reactions. It is also the case of witnessing death of patients as it can be emotionally bleeding while also representing a risk factor for burnout. Taken together, the distressing characteristics of the environment and the exposure to traumatic incidents and death contribute to acute and chronic trauma that can build cumulatively over time [1–3]. It is unsurprising that clinical conditions prevalence, such as depression, anxiety disorder, and post-traumatic stress disorder, among ECCUs, persists despite decades of knowledge and efforts to prevent it [4–7]. Indubitably, reasons for this relate to these health services' working and organizational characteristics.

To prevent clinical conditions of ECCU staff, the literature presents much evidence-based interventions to support such workers. Scholars and practitioners developed professional-oriented evidence-based training interventions following different approaches (i.e., cognitive behavioral therapy, CBT, and mindfulness approach [8], relational supportive [9] and coping mechanisms approach [10]). However, such initiatives often involve best practices for addressing specific clinical conditions (e.g., post-traumatic disorder) or propose only person-oriented strategies to help workers cope with their own experiences. However, the working and organizational context of ECCU can leverage the effectiveness of occasional person-oriented training initiatives [11–14]. Work stress and traumatic events may require the presence of mental-health experts conducting primary prevention programs in place (e.g., debriefing), which may not be present due to the lack of personnel in the unit [15]. Simultaneously, job demands and tasks may reduce the possibility of having the chance and the time to devote attention to the psychological states of the staff [5,16].

According to the literature, working and organizational conditions exposed to work stress and traumatic events may require initiatives to prevent and relational support [3,5,16,17]. Combining the lack of working and personnel resources with the constant exposure to stressful aspects, prevention initiatives can reduce the risk of clinical condition development. This can be the case of training interventions meant to promote psychological knowledge (e.g., psycho-educational interventions) or psychosocial competences and peer-supporting (e.g., defusing) of the staff [11,13,14]. The former follows the evidence that individuals with psychological knowledge can better understand and identify emotions and feelings, especially after traumatic events [13,14]. The latter highlights the importance of building forms of relational support by fostering competences for peer support. As for psycho-education, psychosocial competences can help to act in situ directly after traumatic events and support the elaboration of cognitions, emotions and feelings[14]. However, the potential of such prevention programs and whether they can effectively reduce the development of clinical conditions among the staff are still unclear[13,14]. The literature presents only twenty-three cases of training interventions for peer-supporting [18,19], while differences persist between countries. For example, in the Italian context, while the national regulation recognizes the importance of psychological support for ECCU staff, initiatives are sparse and not formally realized and supported on a national level [20]. Taken together, these aspects highlight the need to better inform the introduction of initiatives to prevent and support ECCU staff, especially in the case of specific structural barriers (e.g., lack of resources).

The present paper reports the results of a mixed methods study meant to evaluate an intervention involving a combination of psycho-educational and defusing trainings. The above insights provided the impetus to conceptualize trauma determinants and management. They suggested a feasible path to address the complex interplay between work stress (e.g., high exposition to stressful and traumatic events) and structural barriers (e.g., lack of resources) to clinical conditions prevention in the ECCU [18,19,21]. Considering the lack of organizational-based initiatives and permanent psychological health programs, our overreaching aim is to advance our empirical knowledge of the potential of prevention programs based on facilitating psychological knowledge and skills to deal with traumatic events in ECCU. Notably, we seek to complement the existing perspectives on pre by analyzing our intervention in order to inform the introduction of mental health prevention programs in the ECCU context. We do so by reporting the results of a mixed methods study. Following the call for the use of mixed methods to evaluate intervention [22], we used such an approach for purposes of (a) expansion (extending breadth and scope) to allow exploration of multiple levels of influence and (b) triangulation to assess the extent to which qualitative and quantitative findings corroborate each other [23]. As such, the present study can help scholars to better understand the effects of a specific prevention program while also support practitioners and healthcare managers in realizing similar initiatives.

[…]

2.2. Evaluating the intervention: Mixed methods

To evaluate our intervention, we conducted a mixed methods project to obtain a retrospective and prospective picture of the state of the trainees and the quality of the training [23] (see Figure 1). Our mixed methods projects is theoretically driven by a deductive approach. As depicted in Figure 1, the first and the second study where represent the retrospective chart, i.e., the psychological status of the staff. In this, quantitative and qualitative data were collected concurrently, but analysed independently. Then, we merged the results and integrated them to create the research narrative and conduct the intervention. The last study was qualitative and represents the prospective component and realized after the intervention in order to capture the experiences and perceptions of effectiveness of the training among the participants.”

RC2: Materials and Methods. 2.1. Psycho-educational-defusing training. There is no article combining psycho-educational and defusing programs (effect on ECCU)?, that aspect is very relevant. The role and relevance of psycho-educational programs is adequately explained. The role and relevance of defusing programs is also well explained. The priority focus of the defusing program is well understood. However, the impact that defusing programs and psycho-educational training can have on ECCU can be improved. In this way it is possible to highlight the content of your article with respect to previous ones. Furthermore, to emphasize the gap in knowledge about these relationships. This section in the methods area is somewhat strange. Reconsider its location in the introduction.

AR2: Thanks for your careful reading of the manuscript. We agree with your points which are also in line with those made by reviewer 3. We provided a more thorough description of the intervention method used. See the following extract:

“2.1. Psycho-educational-defusing training

Surprisingly, although the literature presents possibilities to integrate psycho-educational and defusing programs, there must be evidence of how such a combination could work in ECCU [18,19,21]. First, psycho-educational programs are meant to foster an understanding of the psychological aspects involved in work-related stress and traumatic events. The core idea is to educate the trainees to recognize the psychological valence in their work to foster their capacity to acknowledge their psychological health, risks for their health, and, more broadly, to talk about their psychological suffering. As such, this type of intervention involves transferring essential psychological pieces of knowledge followed by examples from concrete events [13].

Second, a defusing intervention program involves training specialists, i.e., defusers, to manage brief activities after traumatic situations. The defusing intervention follows the basic assumptions of a debriefing session where a mental health professional guide a group of people exposed to a traumatic incident in order to process it. That is, a defusing intervention sees peers trained to guide small groups of people to process traumatic incidents. However, defusers are not trained to perform the same functions as those specialized in mental health and do not replace them. The defusing approach has one main difference from other types of support intervention, namely the duration of the intervention: in fact, the work of the defusers is of short duration, aimed at processing the stressful and traumatic event according to a specific protocol. Moreover, it occurs immediately after the request by the beneficiaries and in situ. A defusing session includes an initial moment of gathering with purposes of a) sharing and bringing out events, feelings, and thoughts related to the traumatic events, b) initiating processing of them (i.e., events, feelings, and thoughts) with c) the peer reassurance[14]. These elements are the object of an intervention aimed at training professionals as defusers and follows both theoretical and practical training.

Ultimately, while the psycho-educational programs offer the basis to open to the psychological dimension at work, training defusers is effective insofar as it aims to offer practical knowledge for on time peer-support interventions after a traumatic event. Psycho-educational and defusing training can work in tandem to help improve both the knowledge and the skills of ECCU.

Following existing evidence-based perspectives while considering possible structural barriers and inevitable work-related stress in ECCU, we propose a unique intervention meant to restore and guarantee the well-being of ECCU staff that can be effective and timely in its realization by de-pathologizing and re-dimensioning traumatic events in situ [9,13,14,21]. We did so by proposing two separate training in on intervention session. While the first part was devoted to psycho-educational training (four hours), the second part aimed at offering the knowledge and practical skills for conducting a defusing session.”

CR3: 2.2. Evaluating the intervention: Mixed methods. Possibly Figure 1 would fit better in this section. The mixed study configuration is well explained. Moreover, it is one of the relevant parts of this article. An explanation of the online questionnaire (process) is needed. What guarantees did the participant have, e.g., confidentiality of data, voluntary withdrawal, consent to participate and so on. This information could be entered in the procedure section because I see that in this section they enter some data. The results of the retrospective component were initially returned to the participants. Why? 2.3.  Participants & Procedure. Have the distribution of the population studied as a possible limitation of your article. 70% female. Remember to contextualize the problem in the introduction and include more information on the five hospitals analyzed.  The qualitative retrospective component is well explained as is the whole process in general. Good job.

AR3: As previously reported, we moved Figure 1 in this section, thanks for your prompt. Moreover, we provided more details about the data collection. You can identify our revisions both in the manuscript text and in the following extract:

“2.3. Participants & Procedure

For the retrospective component concerning the cross-sectional study, 55% of the personnel of the ECCU (N = 404) of the five hospitals involved took part. The questionnaires were administered to 225 healthcare workers, of whom 222 consented to use the data. The target population consisted of 45 registered nurses, 136 nurses, and 41 physicians. The average age of the participants was 43 years (M = 42.6), and there was a prevalence of women corresponding to 70% of the sample (n = 157). Regarding the qualitative retrospective component, 21 participants belonged to the five previously mentioned hospitals. Lastly, 61 participants among the staff were involved in the prospective component. Participation to our project was voluntary and there was not any financial incentive for it.

We began with the cross-sectional study by administrating an online questionnaire on ECCU's staff from five local hospitals. Subsequently, autobiographical stories were collected by asking the same sample to realize an autobiography of their traumatic events. At the end of the retrospective part, we continued with three days of intervention with three different groups of ECCU staff belonging to the cross-sectional study. Each session lasted eight hours and was not included working hours. During which participants were introduced to the notions of clinical conditions such as psychological distress, anxiety, depression, and panic attack disorder and were invited to open up and collectively share their experience (i.e., psycho-educational program). During this session, results narrative of the two retrospective study was returned in order to offer a depiction of the psychological status of the staff. This allowed to offer a concrete basis and examples for realizing the psycho-educational program. After the educational phase, we concluded each session with the defusing program. In this case, we explained defusing and simulated a session of defusing. During the simulation, trainees and participants role-played different possible events that an ECCU employee may witness during their shift (e.g., handling a relative in a crisis of anger). At the end of the simulations impersonated by the participants themselves, they were asked to conduct a defusing session by recounting the events, reporting thoughts and emotions, and trying to console themselves. During the simulations, it was ensured that the activity could be interrupted at any time by using a safe word that participants could use in the face of overly emotional states. Finally, we concluded each meeting by training volunteers to specialize in the defusing stages.

Finally, after the intervention days, we invited participants of the training to take part in follow-up meetings to discuss the quality of the training and their experiences after that (i.e., prospective study, focus groups). That is, two researchers realized three focus groups in which they opened the discussion among participants by asking them to report their experiences and perceptions of the training. No additional specific questions guided the discussion as space was offered for sharing thoughts in an open and inclusive climate. These elements were important for capturing the narratives of the participants whose analysis helped to understand the effectiveness of the intervention.

In all of the phases of our mixed methods study, participants were informed about the aim of the study and instructed about the procedure by the researchers. Informed consent was obtained from all participants whose data are used in the study maintaining their anonymity. Moreover, given the content of the questions, participants could withdraw from each study whenever they wanted, and psychological support was offered upon requests.”

RC4: 2.4. Instruments. Why did they use the Maslach Burnout Inventory (MBI) to measure stress? Possibly there are better and more specific instruments available. Using the Italian version is fine. Using a measurement scale to assess how traumatic events might influence not only the level of stress is fine (Italian version of the Death Attitude Profile-Revised (DAP-R)). The model of Zimmer-Gembeck and Skinner (2011) to identify personal strategies is very good. The way to measure the qualitative part, specifically, the retrospective components is well explained. They can explain in more detail the focus groups and their importance?.

AR4: Thanks for your appreciation for our instruments. We used the MBI measure as it is the most implemented to investigate stress in the healthcare context, as well as one of the well-known measures by non-academic audience. In parallel, its reliability and validity supports its use. With respect to the qualitative component, we provided more details about the focus groups. Please refer to the following extract:

“Finally, after the intervention days, we invited participants of the training to take part in follow-up meetings to discuss the quality of the training and their experiences after that (i.e., prospective study, focus groups). That is, two researchers realized three focus groups in which they opened the discussion among participants by asking them to report their experiences and perceptions of the training. No additional specific questions guided the discussion as space was offered for sharing thoughts in an open and inclusive climate. These elements were important for capturing the narratives of the participants whose analysis helped to understand the effectiveness of the intervention.

In all of the phases of our mixed methods study, participants were informed about the aim of the study and instructed about the procedure by the researchers. Informed consent was obtained from all participants whose data are used in the study maintaining their anonymity. Moreover, given the content of the questions, participants could withdraw from each study whenever they wanted, and psychological support was offered upon requests.”

RC6: 2.5. Analytical strategy. This part is a priority, and the analysis is well focused. It is key to highlight that this is the first study of this nature (mixed). They did not use any software to analyze quantitative data, e.g., Atlas?. 2.6. Ethical considerations. Having the support of an ethics committee is of vital importance and enhances the results obtained. 3.      Results. 3.1. Retrospective: quantitative. It is necessary to better explain the percentages obtained through descriptive statistics. That is, did you use correlations? Please detail all the processes used. Specifying dysfunctional coping strategies is very important. This part is fine. 3.2. Retrospective: qualitative I especially like the way they explain this part. Good job. 3.3. Prospective: quantitative In the end there were six or three focus groups. This part is confusing. In general, the prospective: quantitative is well explained.

AR6: Thanks for your appreciation and your questions. We clarified all these aspects. For example, we did the qualitative analysis manually. We added this information in the manuscript.

“Our study is the first of its nature, so we employed an exploratory approach in our analysis. For the cross-sectional data, we limited to descriptive statistics and correla-tional analysis. Data collected via autobiographies and focus groups were analyzed manually using content analysis.”

RC7: 4.      Discussion. The explanation of the strengths of the applied methodology is good. I agree on the importance of your study and that the mixed methodology is one of its main strengths. However, it is important to better discuss your findings and generate some kind of comparison with previous studies even if partial before proposing future research. Being able to better analyze the results through discussion is key. Their research suggests a greater number of practical applications. Expand on that part. I believe it is one of the main contributions of this research. Check how you measured resilience. Finally, it is essential to include limitations.

AR7: Thanks for your constructive comments. These are in line with the comments by reviewers 1 and 3. We revised the entire discussion and included a) theoretical contributions, b) limitations and future research, and c) the applied implications. Please refer to the following extract of the discussion that we reported for your convenience:

Our mixed-method project found that psycho-educational-defusing interventions can be effective in preventing traumatic events despite the resource limitations of ECCU. The units of emergency and critical care in the healthcare sector inevitably expose their staff to acute stress and traumatic events. Moreover, these units do not always present resources for preventing clinical conditions and promoting the well-being of their team. Considering these barriers, our psycho-educational-defusing intervention appeared as a flexible and practical approach against organizational obstacles. As such, our study responds to the call for realizing empirical investigations of preventing programs [19], while also providing the first evidence of the effectiveness of a combined training [9,13,14,21]. Traumas and grief among ECCUs can likely persist unless psychological knowledge and psychosocial competences are addressed. Results of our project revealed that a) improving knowledge and competences to express feelings and thoughts related to stressful events while b) creating the conditions for peer-supporting maybe even more effective than secondary prevention programs [9,13,14,19,21].

According to the literature, ECCU staff are constantly exposed to potential traumatic incidents with a high risk for their staff for emotional, behavioural and physical reactions and the development of clinical conditions. This study’s results were not an exception and we found similar trends of psychological suffering among ECCU staff due traumatic incidents and distressing conditions [1–7]. In this, trauma is cumulative with a sequela of psychological suffering that affects the overall experience and practice of the staff [3]. That is, trauma and grief appear as emotional, behavioural, and physical reactions that result from the unpredictability, overcrowding, continuous confrontation with a broad range of traumatic incidents. However, this is not only related to the momentary, occasional experience but last overtime which can have a sequela of psychological suffering that impacts the way a) ECCU’s staff are able to overcome distressing conditions and b) their practices. In this, supplying ECCU staff with group resources may be beneficial and create the conditions for empowering ECCU staff [28]. Results of the focus groups revealed that the main contribution of our intervention lays in the provision of psychological knowledge which offers a vocabulary to the staff for verbalizing and sharing their inner thoughts and emotions while also creating opportunities for recognizing their grief. In parallel, peer-support appeared to be a tool for overcoming distressing incident as participants of the focus group reported how they had occasion to reduce their emotional fatigue by engaging in defusing sessions [28,29]. This resulted to be helpful for processing the incident but also to build meaningful relationships. Taken together, these elements indicate that our training while offering a essential resource for ECCU staff, it also support the quality of team work and relationships among peers.

4.1 Limitations & Future research

This study is the first of its kind to examine primary preventing programs via a mixed method project, and as such, our is intended as exploratory. While this can limit its results, it also yields a series of implications for future research. First, our project participants were from a specific context (i.e., the Italian health sector) and specific healthcare sector departments (i.e., ECCU) they may not reflect the broader population. Despite this, our results represent a basis for future exploration extending to other countries and departments. Second, the absence of quantitative evidence of the effectiveness of the study requires careful consideration. Future investigations can consider implementing statistical modeling to reflect the vast experiences of this staff. Longitudinal studies and experimental studies involving control groups can be implemented for providing evidence-based knowledge in support to our findings. Lastly, our qualitative studies involved only a less proportion of the ECCU staff of the hospitals involved and no physicians attended the focus groups. This suggests that there might be an additional number of different meanings that ECCU staff use to talk about trauma and grief. However, the variety of meanings reported indicates that we must also consider with caution any research that might stereotype the lived experience of individuals.

In addition, future studies can also take into account our results independently of our study limitations. Notably, our results suggest several implications for the study of trauma and grief in the healthcare context. First, the fact that we identify a series of structural problems suggests that healthcare organizations should prioritize attention to trauma in the ECCU context. This can result in continuous monitoring of the staff while maintaining and creating occasions for psycho-education. Second, our results of the lived experience of ECCU staff appear to be ideal examples for understanding the disparate and different meanings of trauma and grief. These entail something more than objective traumatic acute stressful incidents. Our results show that trauma is cumulative [3] and that trauma results in repetitive and constant frustration of the staff, which leaves them less willing to express emotions and feelings [1,2,18]. Lastly, the present study adds to the current understanding of recovery from distressing events. While building resilience, developing coping strategies, and seeking professional support are essential keys to recovery, our study emphasizes the importance of community and peer support within the work environment. Soliciting peer support involves recognizing structural conditions while stressing the importance of peer relations and trust as reassuring resources that can foster recovery [28,29].

4.2 Practical implications

Our results provided initial knowledge on the potential of psycho-educational defusing intervention as an effective training program for preventing mental-ill health problems among ECCU staff. Then, our results can help healthcare managers in realizing initiatives for providing psychological support to their staff, and especially in presence of lack of financial resources for continuous support by mental health experts. That is, our study sharpens the specific role of psychological education and peer support. While psycho-education can foster awareness over the psychological effects of trauma and distressing conditions, defusing emphasizes the value of peer supporting by exchanging and sharing thoughts and feeling among colleagues in situ, right after incidents. In this, the main strength is the accomplished effectiveness o f a relatively short and on time intervention.

Realizing initiatives to mitigate the effects of the environment is important to improve the overall working experience of ECCU staff while also promoting the ECCU context itself and the patients for which it cares of. Accordingly, offering a supporting environment with an organizational network offering immediate defusing thanks to peer support [3,15]. In our study results, we showed how ECCU staff are in need for such types of intervention as they can be not only be helpful but also necessary, with the addition of organizational support.

RC8: 5.      Conclusion. The conclusion includes limitations and even future research. I suggest that you reconsider the discussion section by improving the practical implications, future research and including limitations. The conclusion section, in my opinion, should be redrafted through the main objective of this research and its possible fulfillment.

AR8: Thanks a lot, we agree with your point. It is in line with those made by reviewer 3. Please, refer to our new conclusion of the manuscript as follows:

  1. Conclusion

      The present study, which adopted a mixed methods approach, explored the effectiveness of a psycho-educational-defusing intervention for preventing trauma and grief among the ECCU staff. Considering the continuous exposition to distressing conditions and traumatic incidents, the ECCU staff confronts with a series of structural barriers which remind to the need for in situ prevention programs. Our mixed methods study allowed us to find that trauma in the ECCU occurs in a variety of disparate ways and is cumulative. Trauma exerts effects on the mental health, and professional practice while also remaining unprocessed by the staff. Psycho-educational-defusing intervention can be a resource for ECCU departments as results showed that our training represented a protective factor thanks to the offer of knowledge and skills for coping with distress and trauma.

Reviewer 3 Report

Comments and Suggestions for Authors

Author Response

Reviewer 3

RC1: Title: Preventing Trauma and Grief in Emergency and Critical Care Units: A Mixed Methods Study on a Psycho-educational Defus-ing Intervention Thank you for submitting the manuscript for review. This study examines the effectiveness of a combined psychoeducational and defusing intervention for the prevention of trauma and grief in emergency and critical care unit (ECCU) staff. Given the high levels of stress and increased risk of mental illness in these work settings, the study aims to identify preventive interventions and evaluate their effectiveness. Using a mixed-methods approach that includes both quantitative and qualitative data, the study will provide a comprehensive overview of the psychological states of staff and the impact of the intervention. The results provide valuable insights into improving the psychological well-being of ECCU employees and highlight the need for ongoing preventive programs. Overall, the study provides important insights into the prevention of trauma and grief in the ECCU setting. There are large gaps in all chapters that require significant revision.

AR1: We wish to thank you for your careful reading of the manuscript. Your comments and those made by reviewer 1 and reviewer 2 helped us to improve the quality of our manuscript. In the following, we report the way we operated in the manuscript text and our extensive revisions. Again, thank you for your help.

RC2: General comments: 1) Please read the author guidelines before the next submission and adapt the publication accordingly (e.g., affiliations for names with capitalized letters. The contact details are missing. Formatting of headings and subheadings, references). 2) In the text, reference numbers should be placed in square brackets [ ]. 3) Please use an up-to-date template. Does the header show "Healthcare 2021"?

AR2: Thanks for your careful reading of the manuscript. We revised our manuscript according to the up-to-date indications for article submissions to the journal Heatlhcare.

RC3: Introduction: 4) What traumatic events can be expected in the emergency department and ECCU? You are welcome to include some examples in the introduction. 5) Indubitably, reasons for this relate to these health services' working and organizational characteristics. I don't see it that way. Factors such as overcommitment, personality factors, etc. also play a role here. You should rephrase this sentence or back it up with references. 6) The introduction should explain the terms trauma and grief in more detail to better reflect the title of the study. 7) The introduction could benefit from the inclusion of prevalence data on trauma and grief in healthcare professionals. This would emphasise the importance of your work. 8) There is a lack of a clear research question and explicit hypotheses, which are important for the structure and objectives of the study.  9) Figure 1 is also not legible in printed form. Figure 1 is blurred. 10) A chapter should not end with a figure or table.

AR3: Your comments are in line with those made by reviewers 1 and 2. We agree with your points and we revised the entire introduction by a) describing the meaning and prevalence of trauma in the context of ECCU. We b) also included details about the objectives and contributions of our manuscript in order to make the overall contribution clear. Finally, we c) moved Figure 1 in the Method section and we realized a clearer picture. For your convenience, we report our revised version of the introduction below.

“1. Introduction

Trauma and grief are common in sectors that inevitably expose their staff to continuous work stress and traumatic events, particularly in Emergency and Critical Care Units (ECCU) of the health sector. Unpredictability, overcrowding, continuous confrontation with a broad range of patients with different and disparate of disease, while also confronting with time pressures and engagement with a multiplicity of responsibilities are unceasing distressing conditions. These can cause emotional, behavioural and physical reactions. It is also the case of witnessing death of patients as it can be emotionally bleeding while also representing a risk factor for burnout. Taken together, the distressing characteristics of the environment and the exposure to traumatic incidents and death contribute to acute and chronic trauma that can build cumulatively over time [1–3]. It is unsurprising that clinical conditions prevalence, such as depression, anxiety disorder, and post-traumatic stress disorder, among ECCUs, persists despite decades of knowledge and efforts to prevent it [4–7]. Indubitably, reasons for this relate to these health services' working and organizational characteristics.

To prevent clinical conditions of ECCU staff, the literature presents much evidence-based interventions to support such workers. Scholars and practitioners developed professional-oriented evidence-based training interventions following different approaches (i.e., cognitive behavioral therapy, CBT, and mindfulness approach [8], relational supportive [9] and coping mechanisms approach [10]). However, such initiatives often involve best practices for addressing specific clinical conditions (e.g., post-traumatic disorder) or propose only person-oriented strategies to help workers cope with their own experiences. However, the working and organizational context of ECCU can leverage the effectiveness of occasional person-oriented training initiatives [11–14]. Work stress and traumatic events may require the presence of mental-health experts conducting primary prevention programs in place (e.g., debriefing), which may not be present due to the lack of personnel in the unit [15]. Simultaneously, job demands and tasks may reduce the possibility of having the chance and the time to devote attention to the psychological states of the staff [5,16].

According to the literature, working and organizational conditions exposed to work stress and traumatic events may require initiatives to prevent and relational support [3,5,16,17]. Combining the lack of working and personnel resources with the constant exposure to stressful aspects, prevention initiatives can reduce the risk of clinical condition development. This can be the case of training interventions meant to promote psychological knowledge (e.g., psycho-educational interventions) or psychosocial competences and peer-supporting (e.g., defusing) of the staff [11,13,14]. The former follows the evidence that individuals with psychological knowledge can better understand and identify emotions and feelings, especially after traumatic events [13,14]. The latter highlights the importance of building forms of relational support by fostering competences for peer support. As for psycho-education, psychosocial competences can help to act in situ directly after traumatic events and support the elaboration of cognitions, emotions and feelings[14]. However, the potential of such prevention programs and whether they can effectively reduce the development of clinical conditions among the staff are still unclear[13,14]. The literature presents only twenty-three cases of training interventions for peer-supporting [18,19], while differences persist between countries. For example, in the Italian context, while the national regulation recognizes the importance of psychological support for ECCU staff, initiatives are sparse and not formally realized and supported on a national level [20]. Taken together, these aspects highlight the need to better inform the introduction of initiatives to prevent and support ECCU staff, especially in the case of specific structural barriers (e.g., lack of resources).

The present paper reports the results of a mixed methods study meant to evaluate an intervention involving a combination of psycho-educational and defusing trainings. The above insights provided the impetus to conceptualize trauma determinants and management. They suggested a feasible path to address the complex interplay between work stress (e.g., high exposition to stressful and traumatic events) and structural barriers (e.g., lack of resources) to clinical conditions prevention in the ECCU [18,19,21]. Considering the lack of organizational-based initiatives and permanent psychological health programs, our overreaching aim is to advance our empirical knowledge of the potential of prevention programs based on facilitating psychological knowledge and skills to deal with traumatic events in ECCU. Notably, we seek to complement the existing perspectives on pre by analyzing our intervention in order to inform the introduction of mental health prevention programs in the ECCU context. We do so by reporting the results of a mixed methods study. Following the call for the use of mixed methods to evaluate intervention [22], we used such an approach for purposes of (a) expansion (extending breadth and scope) to allow exploration of multiple levels of influence and (b) triangulation to assess the extent to which qualitative and quantitative findings corroborate each other [23]. As such, the present study can help scholars to better understand the effects of a specific prevention program while also support practitioners and healthcare managers in realizing similar initiatives.”

RC4: Methods: 11) Begin the paragraph with participants and procedures.

AR4: Thanks for your suggestion, however since the methodology used is not very common in the literature we preferred to begin our methods by a) describing the methodology used for the intervention and b) the methodological approach for evaluating. This aspect was appreciated by the other reviewers and particularly by reviewer 1. We hope that it is okay for you if we keep the structure as it is.

RC5: 12) A graphical representation of the process is helpful. The number of test subjects should also be shown in this diagram.

AR5: Thanks, we are sorry that the printed version of our picture did not allow you to see that number of participants is already present. We hope that our new version of the picture is clearer than before.

RC6: 13) Include details on how the focus groups were conducted, including the number of groups, composition (e.g., professional groups), facilitation methods, and specific questions or topics discussed. 14) What qualifications did the interviewers have? Was there an interview guide?

AR6: Thanks for your careful reading. We included these details in the description of the focus group. Please, refer to the following extract:

Finally, after the intervention days, we invited participants of the training to take part in follow-up meetings to discuss the quality of the training and their experiences after that (i.e., prospective study, focus groups). That is, two researchers realized three focus groups in which they opened the discussion among participants by asking them to report their experiences and perceptions of the training. No additional specific questions guided the discussion as space was offered for sharing thoughts in an open and inclusive climate. These elements were important for capturing the narratives of the participants whose analysis helped to understand the effectiveness of the intervention.

In all of the phases of our mixed methods study, participants were informed about the aim of the study and instructed about the procedure by the researchers. Informed consent was obtained from all participants whose data are used in the study maintaining their anonymity. Moreover, given the content of the questions, participants could withdraw from each study whenever they wanted, and psychological support was offered upon requests.

RC7: 15) Was a required sample size determined at the beginning? 16) Include detailed information about the statistical methods and tests used. Explain how the data were analyzed to evaluate the effectiveness of the intervention.

AR7: Thanks for your question. The sample sized was not pre-determined, yet we included the number of ECCU staff of the five hospitals in order to report the response rate (see our following response, AR8). Also, we explained our analytical strategy. See the following extract for your convenience:

“Our study is the first of its nature, so we employed an exploratory approach in our analysis. For the cross-sectional data, we limited to descriptive statistics. Data collected via autobiographies and focus groups were analyzed manually using content analysis.”

RC8: 17) Clearly define the inclusion and exclusion criteria for participation in the study. Explain the criteria used to ensure that the sample is representative. 18) How was the sampling carried out?

AR8: As reported in AR7, we included these details as follows:

“For the retrospective component concerning the cross-sectional study, 55% of the personnel of the ECCU (N = 404) of the five hospitals involved took part. The questionnaires were administered to 225 healthcare workers, of whom 222 consented to use the data. The target population consisted of 45 registered nurses, 136 nurses, and 41 physicians. The average age of the participants was 43 years (M = 42.6), and there was a prevalence of women corresponding to 70% of the sample (n = 157). Regarding the qualitative retrospective component, 21 participants belonged to the five previously mentioned hospitals. Lastly, 61 participants among the staff were involved in the prospective component. Participation to our project was voluntary and there was not any financial incentive for it.”

RC9: 19) Were the surveys and interventions carried out during working hours?

AR9: Thanks for your questions. We included these details in our description of the intervention. See the following extract:

“We began with the cross-sectional study by administrating an online questionnaire on ECCU's staff from five local hospitals. Subsequently, autobiographical stories were collected by asking the same sample to realize an autobiography of their traumatic events. At the end of the retrospective part, we continued with three days of intervention with three different groups of ECCU staff belonging to the cross-sectional study. Each session lasted eight hours and was not included working hours. During which participants were introduced to the notions of clinical conditions such as psychological distress, anxiety, depression, and panic attack disorder and were invited to open up and collectively share their experience (i.e., psycho-educational program). During this session, results narrative of the two retrospective study was returned in order to offer a depiction of the psychological status of the staff. This allowed to offer a concrete basis and examples for realizing the psycho-educational program. After the educational phase, we concluded each session with the defusing program. In this case, we explained defusing and simulated a session of defusing. During the simulation, trainees and participants role-played different possible events that an ECCU employee may witness during their shift (e.g., handling a relative in a crisis of anger). At the end of the simulations impersonated by the participants themselves, they were asked to conduct a defusing session by recounting the events, reporting thoughts and emotions, and trying to console themselves. During the simulations, it was ensured that the activity could be interrupted at any time by using a safe word that participants could use in the face of overly emotional states. Finally, we concluded each meeting by training volunteers to specialize in the defusing stages.

Finally, after the intervention days, we invited participants of the training to take part in follow-up meetings to discuss the quality of the training and their experiences after that (i.e., prospective study, focus groups). That is, two researchers realized three focus groups in which they opened the discussion among participants by asking them to report their experiences and perceptions of the training. No additional specific questions guided the discussion as space was offered for sharing thoughts in an open and inclusive climate. These elements were important for capturing the narratives of the participants whose analysis helped to understand the effectiveness of the intervention.

In all of the phases of our mixed methods study, participants were informed about the aim of the study and instructed about the procedure by the researchers. Informed consent was obtained from all participants whose data are used in the study maintaining their anonymity. Moreover, given the content of the questions, participants could withdraw from each study whenever they wanted, and psychological support was offered upon requests.”

RC10: 20) Subchapter 2.2 has a high level of plagiarism. This paragraph should be rewritten.

AR10: Thanks for your attention to details. We controlled our manuscript for plagiarism on Compilatio. Results did not reveal any form of plagiarism in our entire manuscript.

RC11: Results: 21) There is no detailed description of the socio-demographic data of the participants, including occupational groups, gender, age and work experience. This is important in order to better assess the composition of the sample and the generalizability of the results. The data should also be presented in tabular form.

AR11: Thanks for asking. Sample description is present in section 2.3.

RC12: 22) The descriptive statistics are not detailed enough. There are no means, standard deviations, and percentages that would provide a clearer overview of the data.

AR12: Thanks for your comment. We limited to report the percentage for symptoms as this was our intention for the study. The percentage cover both the presence and spread of symptoms in our study’s sample.

RC13: 23) Der r-Koeffizient (Korrelationskoeffizient) ist nicht geeignet, um Unterschiede in den soziodemographischen Daten zu vergleichen. Hier sind andere statistische Tests wie der Chi-Quadrat-Test oder der t-Test geeigneter.

AR13: Thanks for your comment. We agree with you and we deleted the correlations.

RC14: 24) In addition, the most important results from subchapters 3.1, 3.2, and 3.3 should also be presented graphically.

AR14: Thanks for your suggestions. However, we are sorry but we are not able to understand what do you mean with the graphical depiction of these results. Should we integrate them in Figure 1?

RC15: 25) The first paragraph on page 7 appears to be a discussion rather than a presentation of results. WHICH ONE?

AR15: We are sorry, but this comment is not clear to us. Please, let us know if the revised version still presents the same issue.

CR16: 26) What traumas are reported? Can you give a percentage?

AR16: Thanks for asking. However, we did not measure trauma but rather investigated it by using qualitative methodologies.   

CR17: Discussion: 27) The strengths and weaknesses should be presented in a separate sub-chapter. The limitations of the study are only superficially mentioned. A more detailed discussion of the methodological and practical limitations would be important to critically scrutinize the validity of the results. How representative is the data? 28) There is no comparison of the results with those of other studies. This is important to place the results in the context of existing research and to strengthen the validity of the findings. 29) There is a lack of concrete proposals for future research. Although the need for further research is mentioned, specific recommendations are missing. 30) First, the fact that we identify a series of structural problems and higher-thanusual levels of burnout Where can you read about this? You don't compare it with other studies, do you? What is high? What is low? 31) Second, our results of the lived experience of ECCU staff appear to be ideal examples for understanding the disparate and different meanings of trauma and grief. How do you arrive at this statement? You're not comparing with other studies, are you? I can't tell that from your results. 32) These entail something more than objective traumatic acute stressful events that result in repetitive and constant frustration of the staff, which leave them less willing to express emotions and feelings. This is also not shown. 33) Lastly, the present study adds to the current understanding of re-covery from distressing events. While building resilience, developing coping strategies, and seeking professional support are essential keys to recovery, our study emphasizes the importance of community resilience and peer support within the work environment. I cannot understand this? Was this measured? How did you measure resilience? 34) The conclusion should be revised. Weaknesses of the work are listed here again. 35) The discussion has a high level of plagiarism. This paragraph should be rewritten.

AR17: Thanks for your points which are in line with those made by reviewers 1 and 2. We agree with all of your questions and suggestions. Accordingly, we revised the entire discussion. Thanks for your suggestions to make our analysis more critical concerning our limitations and future research perspectives. We have operated in the manuscript text and we revised the entire discussion as follows:

“4. Discussion

In this article, we reported the results of a mixed-method project aimed at evaluating a psycho-educational-defusing intervention for preventing trauma and grief in ECCU's staff. Considering the ambition of the study, we followed the mixed-method approach for (a) expansion to allow exploration of multiple levels of influence and (b) triangulation to assess the extent to which qualitative and quantitative findings corroborate each other [23]. The strength of this methodology stands in the qualitative findings, expanding the understanding and uncovering possible explanations for quantitative findings. This is the case for the autobiographies of traumatic events by which we are able to understand the causes of the persistent clinical conditions among the participants. Moreover, the prospective qualitative data of the focus group allowed us to understand the effectiveness of the training. Indeed, to evaluate the effectiveness of defusing sessions, quantitative methods are limited as defusing sessions occur occasionally and based on individual needs. Likewise, prospective quantitative studies would be limited as aspects that emerged from the qualitative sessions cannot be operationalized. However, our methodology and results may inform subsequent studies in which statistical models can be realized to assess the effectiveness of psycho-educational-defusing interventions.

       Our mixed-method project found that psycho-educational-defusing interventions can be effective in preventing traumatic events despite the resource limitations of ECCU. The units of emergency and critical care in the healthcare sector inevitably expose their staff to acute stress and traumatic events. Moreover, these units do not always present resources for preventing clinical conditions and promoting the well-being of their team. Considering these barriers, our psycho-educational-defusing intervention appeared as a flexible and practical approach against organizational obstacles. As such, our study responds to the call for realizing empirical investigations of preventing programs [19], while also providing the first evidence of the effectiveness of a combined training [9,13,14,21]. Traumas and grief among ECCUs can likely persist unless psychological knowledge and psychosocial competences are addressed. Results of our project revealed that a) improving knowledge and competences to express feelings and thoughts related to stressful events while b) creating the conditions for peer-supporting maybe even more effective than secondary prevention programs [9,13,14,19,21].

According to the literature, ECCU staff are constantly exposed to potential traumatic incidents with a high risk for their staff for emotional, behavioural and physical reactions and the development of clinical conditions. This study’s results were not an exception and we found similar trends of psychological suffering among ECCU staff due traumatic incidents and distressing conditions [1–7]. In this, trauma is cumulative with a sequela of psychological suffering that affects the overall experience and practice of the staff [3]. That is, trauma and grief appear as emotional, behavioural, and physical reactions that result from the unpredictability, overcrowding, continuous confrontation with a broad range of traumatic incidents. However, this is not only related to the momentary, occasional experience but last overtime which can have a sequela of psychological suffering that impacts the way a) ECCU’s staff are able to overcome distressing conditions and b) their practices. In this, supplying ECCU staff with group resources may be beneficial and create the conditions for empowering ECCU staff [28]. Results of the focus groups revealed that the main contribution of our intervention lays in the provision of psychological knowledge which offers a vocabulary to the staff for verbalizing and sharing their inner thoughts and emotions while also creating opportunities for recognizing their grief. In parallel, peer-support appeared to be a tool for overcoming distressing incident as participants of the focus group reported how they had occasion to reduce their emotional fatigue by engaging in defusing sessions [28,29]. This resulted to be helpful for processing the incident but also to build meaningful relationships. Taken together, these elements indicate that our training while offering a essential resource for ECCU staff, it also support the quality of team work and relationships among peers.

4.1 Limitations & Future research

This study is the first of its kind to examine primary preventing programs via a mixed method project, and as such, our is intended as exploratory. While this can limit its results, it also yields a series of implications for future research. First, our project participants were from a specific context (i.e., the Italian health sector) and specific healthcare sector departments (i.e., ECCU) they may not reflect the broader population. Despite this, our results represent a basis for future exploration extending to other countries and departments. Second, the absence of quantitative evidence of the effectiveness of the study requires careful consideration. Future investigations can consider implementing statistical modeling to reflect the vast experiences of this staff. Longitudinal studies and experimental studies involving control groups can be implemented for providing evidence-based knowledge in support to our findings. Lastly, our qualitative studies involved only a less proportion of the ECCU staff of the hospitals involved and no physicians attended the focus groups. This suggests that there might be an additional number of different meanings that ECCU staff use to talk about trauma and grief. However, the variety of meanings reported indicates that we must also consider with caution any research that might stereotype the lived experience of individuals.

In addition, future studies can also take into account our results independently of our study limitations. Notably, our results suggest several implications for the study of trauma and grief in the healthcare context. First, the fact that we identify a series of structural problems suggests that healthcare organizations should prioritize attention to trauma in the ECCU context. This can result in continuous monitoring of the staff while maintaining and creating occasions for psycho-education. Second, our results of the lived experience of ECCU staff appear to be ideal examples for understanding the disparate and different meanings of trauma and grief. These entail something more than objective traumatic acute stressful incidents. Our results show that trauma is cumulative [3] and that trauma results in repetitive and constant frustration of the staff, which leaves them less willing to express emotions and feelings [1,2,18]. Lastly, the present study adds to the current understanding of recovery from distressing events. While building resilience, developing coping strategies, and seeking professional support are essential keys to recovery, our study emphasizes the importance of community and peer support within the work environment. Soliciting peer support involves recognizing structural conditions while stressing the importance of peer relations and trust as reassuring resources that can foster recovery [28,29].

4.2 Practical implications

Our results provided initial knowledge on the potential of psycho-educational defusing intervention as an effective training program for preventing mental-ill health problems among ECCU staff. Then, our results can help healthcare managers in realizing initiatives for providing psychological support to their staff, and especially in presence of lack of financial resources for continuous support by mental health experts. That is, our study sharpens the specific role of psychological education and peer support. While psycho-education can foster awareness over the psychological effects of trauma and distressing conditions, defusing emphasizes the value of peer supporting by exchanging and sharing thoughts and feeling among colleagues in situ, right after incidents. In this, the main strength is the accomplished effectiveness o f a relatively short and on time intervention.

Realizing initiatives to mitigate the effects of the environment is important to improve the overall working experience of ECCU staff while also promoting the ECCU context itself and the patients for which it cares of. Accordingly, offering a supporting environment with an organizational network offering immediate defusing thanks to peer support [3,15]. In our study results, we showed how ECCU staff are in need for such types of intervention as they can be not only be helpful but also necessary, with the addition of organizational support.”

CR18: Conclusion: 36) The conclusion has a high level of plagiarism. This paragraph should be rewritten.

AR18: Thanks for your comment. Due to changes in the discussion section, we re-wrote the entire conclusion of our manuscript. We did so also to be more explicit about our contributions. Please, refer to the following extract:

“5. Conclusion

               The present study, which adopted a mixed methods approach, explored the effectiveness of a psycho-educational-defusing intervention for preventing trauma and grief among the ECCU staff. Considering the continuous exposition to distressing conditions and traumatic incidents, the ECCU staff confronts with a series of structural barriers which remind to the need for in situ prevention programs. Our mixed methods study allowed us to find that trauma in the ECCU occurs in a variety of disparate ways and is cumulative. Trauma exerts effects on the mental health, and professional practice while also remaining unprocessed by the staff. Psycho-educational-defusing intervention can be a resource for ECCU departments as results showed that our training represented a protective factor thanks to the offer of knowledge and skills for coping with distress and trauma.”

Round 2

Reviewer 1 Report

Comments and Suggestions for Authors

The improvements made by the authors are notable and have greatly improved the quality of the article.

Author Response

RC1: The improvements made by the authors are notable and have greatly improved the quality of the article.

AR1: Thank you very much for your appreciation; we are happy that you enjoyed our revisions and work on the manuscript. Your comments have substantially helped us to improve the manuscript.

Reviewer 2 Report

Comments and Suggestions for Authors

Dear authors,

Very good work. All suggestions have been addressed in a detailed and correct way.

Thank you very much.

Author Response

RC1: 

Dear authors,

Very good work. All suggestions have been addressed in a detailed and correct way.

Thank you very much.

AR1: Thank you very much for your appreciation; we are happy you enjoyed our revisions and work on the manuscript. Your comments have helped us to improve the manuscript substantially.

Reviewer 3 Report

Comments and Suggestions for Authors

Many thanks for the revision. I don't see a sufficient point-by-point answer.

You have summarised many questions, but I have noticed that some questions/comments remain unanswered. Therefore, I would like to ask you once again to revise the manuscript and provide precise details (including line numbers) in each case. 

The questions are as follows:

RC3: Introduction: 4) What traumatic events can be expected in the emergency department and ECCU? You are welcome to include some examples in the introduction. 5) Indubitably, reasons for this relate to these health services' working and organizational characteristics. I don't see it that way. Factors such as overcommitment, personality factors, etc. also play a role here. You should rephrase this sentence or back it up with references. 6) The introduction should explain the terms trauma and grief in more detail to better reflect the title of the study. 7) The introduction could benefit from the inclusion of prevalence data on trauma and grief in healthcare professionals. This would emphasise the importance of your work. 8) There is a lack of a clear research question and explicit hypotheses, which are important for the structure and objectives of the study. 9) Figure 1 is also not legible in printed form. Figure 1 is blurred. 10) A chapter should not end with a figure or table.

17) Clearly define the inclusion and exclusion criteria for participation in the study. 

CR17: Discussion: 27) The strengths and weaknesses should be presented in a separate sub-chapter. The limitations of the study are only superficially mentioned. A more detailed discussion of the methodological and practical limitations would be important to critically scrutinize the validity of the results. How representative is the data? 28) There is no comparison of the results with those of other studies. This is important to place the results in the context of existing research and to strengthen the validity of the findings. 29) There is a lack of concrete proposals for future research. Although the need for further research is mentioned, specific recommendations are missing. 30) First, the fact that we identify a series of structural problems and higher-thanusual levels of burnout Where can you read about this? You don't compare it with other studies, do you? What is high? What is low? 31) Second, our results of the lived experience of ECCU staff appear to be ideal examples for understanding the disparate and different meanings of trauma and grief. How do you arrive at this statement? You're not comparing with other studies, are you? I can't tell that from your results. 32) These entail something more than objective traumatic acute stressful events that result in repetitive and constant frustration of the staff, which leave them less willing to express emotions and feelings. This is also not shown. 33) Lastly, the present study adds to the current understanding of re-covery from distressing events. While building resilience, developing coping strategies, and seeking professional support are essential keys to recovery, our study emphasizes the importance of community resilience and peer support within the work environment. I cannot understand this? Was this measured? How did you measure resilience? 34) The conclusion should be revised. Weaknesses of the work are listed here again. 35) The discussion has a high level of plagiarism. This paragraph should be rewritten.

The manuscript is incomplete. There is only one illustration? Please insert all figures and tables at the appropriate place in the manuscript. 

Author Response

Reviewer 1

RC1: Many thanks for the revision. I don't see a sufficient point-by-point answer.

You have summarised many questions, but I have noticed that some questions/comments remain unanswered. Therefore, I would like to ask you once again to revise the manuscript and provide precise details (including line numbers) in each case.

The questions are as follows:

AR1: We wish to thank you for your appreciation. Please, apologize for our decision to integrate your questions. We addressed all of your points and we thank you for giving us the opportunity to rework and provide you with a point-by-point answer. See below our responses.

RC2: Introduction: 4) What traumatic events can be expected in the emergency department and ECCU? You are welcome to include some examples in the introduction.

AR2: Thanks for asking. This comment was in line with one point made by reviewer 2. We added some examples in the initial part of the introduction. Please, refer to the following extract:

“Trauma and grief are common in sectors that inevitably expose their staff to continuous work stress and traumatic events, particularly in Emergency and Critical Care Units (ECCU) of the health sector. Unpredictability, overcrowding, continuous confrontation with a broad range of patients with different and disparate of disease, while also confronting with time pressures and engagement with a multiplicity of responsibilities are unceasing distressing conditions. These can cause emotional, behavioural and physical reactions. It is also the case of witnessing death of patients as it can be emotionally bleeding while also representing a risk factor for burnout. Taken together, the distressing characteristics of the environment and the exposure to traumatic incidents and death contribute to acute and chronic trauma that can build cumulatively over time [1–3]. It is unsurprising that clinical conditions prevalence, such as depression, anxiety disorder, and post-traumatic stress disorder, among ECCUs, persists despite decades of knowledge and efforts to prevent it. By way of an example, meta-analytic investigations reported that one out of four nurses are at risk of development of clinical condition, with similar trends before and after the pandemic [4–7]. Indubitably, reasons for this relate to these health services' working and organizational characteristics.”

RC3: 5) Indubitably, reasons for this relate to these health services' working and organizational characteristics. I don't see it that way. Factors such as overcommitment, personality factors, etc. also play a role here. You should rephrase this sentence or back it up with references.

AR3: You are right, there are some individual-level dimensions which can play a role. However, according to the literature in the healthcare sector and organizational psychology, the specific context of ECCU plays a substantial role as it exposes to continuous stressing conditions. We emphasized this aspect in the introduction by including some references. Please refer to the following extract:

“Trauma and grief are common in sectors that inevitably expose their staff to continuous work stress and traumatic events, particularly in Emergency and Critical Care Units (ECCU) of the health sector. Unpredictability, overcrowding, continuous confrontation with a broad range of patients with different and disparate of disease, while also confronting with time pressures and engagement with a multiplicity of responsibilities are unceasing distressing conditions. These can cause emotional, behavioural and physical reactions. It is also the case of witnessing death of patients as it can be emotionally bleeding while also representing a risk factor for burnout. Taken together, the distressing characteristics of the environment and the exposure to traumatic incidents and death contribute to acute and chronic trauma that can build cumulatively over time [1–3]. It is unsurprising that clinical conditions prevalence, such as depression, anxiety disorder, and post-traumatic stress disorder, among ECCUs, persists despite decades of knowledge and efforts to prevent it. By way of an example, meta-analytic investigations reported that one out of four nurses are at risk of development of clinical condition, with similar trends before and after the pandemic [4–7]. Indubitably, reasons for this relate to these health services' working and organizational characteristics.

To prevent clinical conditions of ECCU staff, the literature presents much evidence-based interventions to support such workers. Scholars and practitioners developed professional-oriented evidence-based training interventions following different approaches (i.e., cognitive behavioral therapy, CBT, and mindfulness approach [8], relational supportive [9] and coping mechanisms approach [10]). However, such initiatives often involve best practices for addressing specific clinical conditions (e.g., post-traumatic disorder) or propose only person-oriented strategies to help workers cope with their own experiences. However, the working and organizational context of ECCU can leverage the effectiveness of occasional person-oriented training initiatives [11–14]. Work stress and traumatic events may require the presence of mental-health experts conducting primary prevention programs in place (e.g., debriefing), which may not be present due to the lack of personnel in the unit [15]. Simultaneously, job demands and tasks may reduce the possibility of having the chance and the time to devote attention to the psychological states of the staff [5,16].”

RC4: 6) The introduction should explain the terms trauma and grief in more detail to better reflect the title of the study.

AR4: Thanks for remarking this aspect, which is in line with your previous comment “RC2”. We integrated the two points (RC2 and RC4) and included these elements in the introduction as follows:

“Trauma and grief are common in sectors that inevitably expose their staff to continuous work stress and traumatic events, particularly in Emergency and Critical Care Units (ECCU) of the health sector. Unpredictability, overcrowding, continuous confrontation with a broad range of patients with different and disparate of disease, while also confronting with time pressures and engagement with a multiplicity of responsibilities are unceasing distressing conditions. These can cause emotional, behavioural and physical reactions. It is also the case of witnessing death of patients as it can be emotionally bleeding while also representing a risk factor for burnout. Taken together, the distressing characteristics of the environment and the exposure to traumatic incidents and death contribute to acute and chronic trauma that can build cumulatively over time [1–3].”

RC5: 7) The introduction could benefit from the inclusion of prevalence data on trauma and grief in healthcare professionals. This would emphasise the importance of your work.

AR5: Thanks for asking to include the prevalence of such symptoms and mental-ill health among ECCU staff. Currently, there is a variety of sources, then we decided to include the results of meta-analysis in order to summarize their presence. Please, refer to the following extract:

“It is unsurprising that clinical conditions prevalence, such as depression, anxiety disorder, and post-traumatic stress disorder, among ECCUs, persists despite decades of knowledge and efforts to prevent it. By way of an example, meta-analytic investigations reported that one out of four nurses are at risk of development of clinical condition, with similar trends before and after the pandemic [4–7]. Indubitably, reasons for this relate to these health services' working and organizational characteristics.”

RC6: 8) There is a lack of a clear research question and explicit hypotheses, which are important for the structure and objectives of the study.

AR6: We strongly empathize with your point. We agree with you and we revised our introduction accordingly. We included our objectives and main purposes in order to explain the point of our manuscript. Please, refer to the following extract:

“The literature presents only twenty-three cases of training interventions for peer-supporting [18,19], while differences persist between countries. For example, in the Italian context, while the national regulation recognizes the importance of psy-chological support for ECCU staff, initiatives are sparse and not formally realized and supported on a national level [20]. Taken together, these aspects highlight the need to better inform the introduction of initiatives to prevent and support ECCU staff, espe-cially in the case of specific structural barriers (e.g., lack of resources).

The present paper reports the results of a mixed methods study meant to evaluate an intervention involving a combination of psycho-educational and defusing trainings. The above insights provided the impetus to conceptualize trauma determinants and management. They suggested a feasible path to address the complex interplay between work stress (e.g., high exposition to stressful and traumatic events) and structural bar-riers (e.g., lack of resources) to clinical conditions prevention in the ECCU [18,19,21]. Considering the lack of organizational-based initiatives and permanent psychological health programs, our overreaching aim is to advance our empirical knowledge of the potential of prevention programs based on facilitating psychological knowledge and skills to deal with traumatic events in ECCU. Notably, we seek to complement the ex-isting perspectives on pre by analyzing our intervention in order to inform the intro-duction of mental health prevention programs in the ECCU context. We do so by re-porting the results of a mixed methods study. Following the call for the use of mixed methods to evaluate intervention [22], we used such an approach for purposes of (a) expansion (extending breadth and scope) to allow exploration of multiple levels of in-fluence and (b) triangulation to assess the extent to which qualitative and quantitative findings corroborate each other [23]. As such, the present study can help scholars to better understand the effects of a specific prevention program while also support prac-titioners and healthcare managers in realizing similar initiatives.”

RC7: 9) Figure 1 is also not legible in printed form. Figure 1 is blurred.

AR7: Thanks for letting us know about the figure. We revised it and make it legible. Please, let us know how it works now.

RC8: 10) A chapter should not end with a figure or table.

AR8: We agree with you, and we moved our figure to the following section.

RC9: 17) Clearly define the inclusion and exclusion criteria for participation in the study.

AR9: Thanks for asking. We included this aspect in the Procedure and Participants section as follows:

“For the retrospective component concerning the cross-sectional study, 55% of the personnel of the ECCU (N = 404) of the five hospitals involved took part. The questionnaires were administered to 225 healthcare workers, of whom 222 consented to use the data. The target population consisted of 45 registered nurses, 136 nurses, and 41 physicians. The average age of the participants was 43 years (M = 42.6), and there was a prevalence of women corresponding to 70% of the sample (n = 157). Regarding the qualitative retrospective component, 21 participants belonged to the five previously mentioned hospitals. Lastly, 61 participants among the staff were involved in the prospective component. Participation to our project was voluntary and there was not any financial incentive for it.”

CR10: Discussion: 27) The strengths and weaknesses should be presented in a separate sub-chapter. The limitations of the study are only superficially mentioned. A more detailed discussion of the methodological and practical limitations would be important to critically scrutinize the validity of the results. How representative is the data?

AR10: Thanks for giving us indications on how to improve the discussion section of our manuscript. We followed your points and we added a) one specific sub-paragraph for theoretical contributions (in which we discussed how our results are in line with existing perspectives), b) one on limitations and future research perspectives and c) one on the applied implications. Please, refer to the following extract:

Discussion

[…]

According to the literature, ECCU staff are constantly exposed to potential traumatic incidents with a high risk for their staff for emotional, behavioural and physical reactions and the development of clinical conditions. This study’s results were not an exception and we found similar trends of psychological suffering among ECCU staff due traumatic incidents and distressing conditions [1–7]. In this, trauma is cumulative with a sequela of psychological suffering that affects the overall experience and practice of the staff [3]. That is, trauma and grief appear as emotional, behavioural, and physical reactions that result from the unpredictability, overcrowding, continuous confrontation with a broad range of traumatic incidents. However, this is not only related to the momentary, occasional experience but last overtime which can have a sequela of psychological suffering that impacts the way a) ECCU’s staff are able to overcome distressing conditions and b) their practices. In this, supplying ECCU staff with group resources may be beneficial and create the conditions for empowering ECCU staff [28]. Results of the focus groups revealed that the main contribution of our intervention lays in the provision of psychological knowledge which offers a vocabulary to the staff for verbalizing and sharing their inner thoughts and emotions while also creating opportunities for recognizing their grief. In parallel, peer-support appeared to be a tool for overcoming distressing incident as participants of the focus group reported how they had occasion to reduce their emotional fatigue by engaging in defusing sessions [28,29]. This resulted to be helpful for processing the incident but also to build meaningful relationships. Taken together, these elements indicate that our training while offering a essential resource for ECCU staff, it also support the quality of team work and relationships among peers.

4.1 Limitations & Future research

This study is the first of its kind to examine primary preventing programs via a mixed method project, and as such, our is intended as exploratory. While this can limit its results, it also yields a series of implications for future research. First, our project participants were from a specific context (i.e., the Italian health sector) and specific healthcare sector departments (i.e., ECCU) they may not reflect the broader population. Despite this, our results represent a basis for future exploration extending to other countries and departments. Second, the absence of quantitative evidence of the effectiveness of the study requires careful consideration. Future investigations can consider implementing statistical modeling to reflect the vast experiences of this staff. Longitudinal studies and experimental studies involving control groups can be implemented for providing evidence-based knowledge in support to our findings. Lastly, our qualitative studies involved only a less proportion of the ECCU staff of the hospitals involved and no physicians attended the focus groups. This suggests that there might be an additional number of different meanings that ECCU staff use to talk about trauma and grief. However, the variety of meanings reported indicates that we must also consider with caution any research that might stereotype the lived experience of individuals.

In addition, future studies can also take into account our results independently of our study limitations. Notably, our results suggest several implications for the study of trauma and grief in the healthcare context. First, the fact that we identify a series of structural problems suggests that healthcare organizations should prioritize attention to trauma in the ECCU context. This can result in continuous monitoring of the staff while maintaining and creating occasions for psycho-education. Second, our results of the lived experience of ECCU staff appear to be ideal examples for understanding the disparate and different meanings of trauma and grief. These entail something more than objective traumatic acute stressful incidents. Our results show that trauma is cumulative [3] and that trauma results in repetitive and constant frustration of the staff, which leaves them less willing to express emotions and feelings [1,2,18]. Lastly, the present study adds to the current understanding of recovery from distressing events. While building resilience, developing coping strategies, and seeking professional support are essential keys to recovery, our study emphasizes the importance of community and peer support within the work environment. Soliciting peer support involves recognizing structural conditions while stressing the importance of peer relations and trust as reassuring resources that can foster recovery [28,29].

4.2 Practical implications

Our results provided initial knowledge on the potential of psycho-educational defusing intervention as an effective training program for preventing mental-ill health problems among ECCU staff. Then, our results can help healthcare managers in realizing initiatives for providing psychological support to their staff, and especially in presence of lack of financial resources for continuous support by mental health experts. That is, our study sharpens the specific role of psychological education and peer support. While psycho-education can foster awareness over the psychological effects of trauma and distressing conditions, defusing emphasizes the value of peer supporting by exchanging and sharing thoughts and feeling among colleagues in situ, right after incidents. In this, the main strength is the accomplished effectiveness o f a relatively short and on time intervention.

Realizing initiatives to mitigate the effects of the environment is important to improve the overall working experience of ECCU staff while also promoting the ECCU context itself and the patients for which it cares of. Accordingly, offering a supporting environment with an organizational network offering immediate defusing thanks to peer support [3,15]. In our study results, we showed how ECCU staff are in need for such types of intervention as they can be not only be helpful but also necessary, with the addition of organizational support.

CR11: 28) There is no comparison of the results with those of other studies. This is important to place the results in the context of existing research and to strengthen the validity of the findings.

AR11: Thanks for asking, this point is in line with the previous one (i.e., CR10). Please refer to our previous response (AR10) and the following extract:

According to the literature, ECCU staff are constantly exposed to potential traumatic incidents with a high risk for their staff for emotional, behavioural and physical reactions and the development of clinical conditions. This study’s results were not an exception and we found similar trends of psychological suffering among ECCU staff due traumatic incidents and distressing conditions [1–7]. In this, trauma is cumulative with a sequela of psychological suffering that affects the overall experience and practice of the staff [3]. That is, trauma and grief appear as emotional, behavioural, and physical reactions that result from the unpredictability, overcrowding, continuous confrontation with a broad range of traumatic incidents. However, this is not only related to the momentary, occasional experience but last overtime which can have a sequela of psychological suffering that impacts the way a) ECCU’s staff are able to overcome distressing conditions and b) their practices. In this, supplying ECCU staff with group resources may be beneficial and create the conditions for empowering ECCU staff [28]. Results of the focus groups revealed that the main contribution of our intervention lays in the provision of psychological knowledge which offers a vocabulary to the staff for verbalizing and sharing their inner thoughts and emotions while also creating opportunities for recognizing their grief. In parallel, peer-support appeared to be a tool for overcoming distressing incident as participants of the focus group reported how they had occasion to reduce their emotional fatigue by engaging in defusing sessions [28,29]. This resulted to be helpful for processing the incident but also to build meaningful relationships. Taken together, these elements indicate that our training while offering a essential resource for ECCU staff, it also support the quality of team work and relationships among peers.

CR12: 29) There is a lack of concrete proposals for future research. Although the need for further research is mentioned, specific recommendations are missing.

AR12: We agree with your point. Thanks for asking to integrate our discussion with research implications. See the following extract:

“4.1 Limitations & Future research

This study is the first of its kind to examine primary preventing programs via a mixed method project, and as such, our is intended as exploratory. While this can limit its results, it also yields a series of implications for future research. First, our project participants were from a specific context (i.e., the Italian health sector) and specific healthcare sector departments (i.e., ECCU) they may not reflect the broader population. Despite this, our results represent a basis for future exploration extending to other countries and departments. Second, the absence of quantitative evidence of the effectiveness of the study requires careful consideration. Future investigations can consider implementing statistical modeling to reflect the vast experiences of this staff. Longitudinal studies and experimental studies involving control groups can be implemented for providing evidence-based knowledge in support to our findings. Lastly, our qualitative studies involved only a less proportion of the ECCU staff of the hospitals involved and no physicians attended the focus groups. This suggests that there might be an additional number of different meanings that ECCU staff use to talk about trauma and grief. However, the variety of meanings reported indicates that we must also consider with caution any research that might stereotype the lived experience of individuals.

In addition, future studies can also take into account our results independently of our study limitations. Notably, our results suggest several implications for the study of trauma and grief in the healthcare context. First, the fact that we identify a series of structural problems suggests that healthcare organizations should prioritize attention to trauma in the ECCU context. This can result in continuous monitoring of the staff while maintaining and creating occasions for psycho-education. Second, our results of the lived experience of ECCU staff appear to be ideal examples for understanding the disparate and different meanings of trauma and grief. These entail something more than objective traumatic acute stressful incidents. Our results show that trauma is cumulative [3] and that trauma results in repetitive and constant frustration of the staff, which leaves them less willing to express emotions and feelings [1,2,18]. Lastly, the present study adds to the current understanding of recovery from distressing events. While building resilience, developing coping strategies, and seeking professional support are essential keys to recovery, our study emphasizes the importance of community and peer support within the work environment. Soliciting peer support involves recognizing structural conditions while stressing the importance of peer relations and trust as reassuring resources that can foster recovery [28,29].”

CR13: 30) First, the fact that we identify a series of structural problems and higher-thanusual levels of burnout Where can you read about this? You don't compare it with other studies, do you? What is high? What is low?

AR13: Thanks for your careful reading of the manuscript. These sentences appeared to be ambiguous and we decided to delete them in order to avoid confusion. We really thanks for making us to notice it.

CR14: 31) Second, our results of the lived experience of ECCU staff appear to be ideal examples for understanding the disparate and different meanings of trauma and grief. How do you arrive at this statement? You're not comparing with other studies, are you? I can't tell that from your results.

AR14: Thanks also for this comment. You are right. We did not make any comparisions. Thanks to your indications, we integrated this aspect with current evidence and perspectives on trauma. Please, refer to the following extract for your convenience:

Second, our results of the lived experience of ECCU staff appear to be ideal examples for understanding the disparate and different meanings of trauma and grief. These entail something more than objective traumatic acute stressful incidents. Our results show that trauma is cumulative [3] and that trauma results in repetitive and constant frustration of the staff, which leaves them less willing to express emotions and feelings [1,2,18]. Lastly, the present study adds to the current understanding of recovery from distressing events. While building resilience, developing coping strategies, and seeking professional support are essential keys to recovery, our study emphasizes the importance of community and peer support within the work environment. Soliciting peer support involves recognizing structural conditions while stressing the importance of peer relations and trust as reassuring resources that can foster recovery [28,29].

CR15: 32) These entail something more than objective traumatic acute stressful events that result in repetitive and constant frustration of the staff, which leave them less willing to express emotions and feelings. This is also not shown.

AR15: Thanks for reporting this element. Indeed, our previous version was not clear enough. Please refer to the new section (which integrates our revisions made according to CR14). See the following extract:

Second, our results of the lived experience of ECCU staff appear to be ideal examples for understanding the disparate and different meanings of trauma and grief. These entail something more than objective traumatic acute stressful incidents. Our results show that trauma is cumulative [3] and that trauma results in repetitive and constant frustration of the staff, which leaves them less willing to express emotions and feelings [1,2,18]. Lastly, the present study adds to the current understanding of recovery from distressing events. While building resilience, developing coping strategies, and seeking professional support are essential keys to recovery, our study emphasizes the importance of community and peer support within the work environment. Soliciting peer support involves recognizing structural conditions while stressing the importance of peer relations and trust as reassuring resources that can foster recovery [28,29].

CR16: 33) Lastly, the present study adds to the current understanding of re-covery from distressing events. While building resilience, developing coping strategies, and seeking professional support are essential keys to recovery, our study emphasizes the importance of community resilience and peer support within the work environment. I cannot understand this? Was this measured? How did you measure resilience?

AR16: This point is in line with one made by Reviewer 2. We agree with both of you. Speaking about resilience was confusing as we did not measure it. We removed these elements from our version.

CR17: 34) The conclusion should be revised. Weaknesses of the work are listed here again.

AR17: Thanks a lot for your comment. We rewrote the Conclusion section as follows:

“The present study, which adopted a mixed methods approach, explored the effectiveness of a psycho-educational-defusing intervention for preventing trauma and grief among the ECCU staff. Considering the continuous exposition to distressing conditions and traumatic incidents, the ECCU staff confronts with a series of structural barriers which remind to the need for in situ prevention programs. Our mixed methods study allowed us to find that trauma in the ECCU occurs in a variety of disparate ways and is cumulative. Trauma exerts effects on the mental health, and professional practice while also remaining unprocessed by the staff. Psycho-educational-defusing intervention can be a resource for ECCU departments as results showed that our training represented a protective factor thanks to the offer of knowledge and skills for coping with distress and trauma.”

CR18: 35) The discussion has a high level of plagiarism. This paragraph should be rewritten.

AR19: Thanks to your comment we completely revised the entire discussion, and we controlled for potential plagiarism using “Compilatio” application. Please, refer to the manuscript text.

CR19: The manuscript is incomplete. There is only one illustration? Please insert all figures and tables at the appropriate place in the manuscript.

AR19: Yes, there is only one illustration. To save space, we opted to limit to only one Figure in the manuscript text.